# Identification of CD133[+] intercellsomes in intercellular communication to offset intracellular signal deficit

Kota Kaneko[1], Yan Liang[1], Qing Liu[1], Shuo Zhang[1], Alexander Scheiter[1,2], Dan Song[1], Gen-Sheng Feng[1]*

[1]Department of Pathology, Department of Molecular Biology, and Moores Cancer Center, University of California at San Diego, La Jolla, United States; [2]Institute of Pathology, University of Regensburg, Regensburg, Germany

**Abstract** CD133 (prominin 1) is widely viewed as a cancer stem cell marker in association with drug resistance and cancer recurrence. Herein, we report that with impaired RTK-Shp2-Ras-Erk signaling, heterogenous hepatocytes form clusters that manage to divide during mouse liver regeneration. These hepatocytes are characterized by upregulated CD133 while negative for other progenitor cell markers. Pharmaceutical inhibition of proliferative signaling also induced CD133 expression in various cancer cell types from multiple animal species, suggesting an inherent and common mechanism of stress response. Super-resolution and electron microscopy localize CD133 on intracellular vesicles that apparently migrate between cells, which we name 'intercellsome.' Isolated CD133[+] intercellsomes are enriched with mRNAs rather than miRNAs. Single-cell RNA sequencing reveals lower intracellular diversity (entropy) of mitogenic mRNAs in Shp2-deficient cells, which may be remedied by intercellular mRNA exchanges between CD133[+] cells. CD133-deficient cells are more sensitive to proliferative signal inhibition in livers and intestinal organoids. These data suggest a mechanism of intercellular communication to compensate for intracellular signal deficit in various cell types.

***For correspondence:**
gfeng@ucsd.edu

**Competing interest:** The authors declare that no competing interests exist.

## eLife assessment

This **important** study was designed to examine the bypass of Ras/Erk signaling defects that enable limited regeneration in a mouse model of hepatic regeneration. This hepatocyte proliferation is associated with the expression by groups of cells of mRNA-loaded CD133+ intracellular vesicles that mediate an intercellular signaling pathway that supports proliferation. These are new observations, supported by **convincing** data, that have broad significance to the fields of regeneration and cancer.

## Introduction

Liver regeneration is a widely used animal model to dissect mechanisms that drive cell proliferation in vivo, with the unique regenerative capacity conserved from animals to humans (***Fausto et al., 2006***; ***Michalopoulos and DeFrances, 1997***; ***Taub, 2004***). One frequently used experimental platform is partial hepatectomy (PHx) of two-thirds liver in rodents. Although re-entering the cell cycle of quiescent hepatocytes in the adult liver is believed to be a primary mechanism of liver regeneration, the possible involvement of stem/progenitor cells has also been explored (***Michalopoulos, 2017***; ***Miyajima et al., 2014***). However, conflicting data exist, regarding the source and contribution of putative stem/progenitor cells in liver regeneration and hepato-carcinogenesis following hepatic injuries (***Kopp et al., 2016***; ***Michalopoulos, 2017***; ***Miyajima et al., 2014***). A CD133[+] cell population was

**eLife digest** The liver is an important metabolic organ that is responsible for digesting nutrients. Over time, it can become damaged by the toxins it receives from food and drink, as well as during infections. Thankfully, cells in the liver can divide and replace the parts that have become harmed allowing the organ to continue carrying out its vital role in the body.

Experiments in mice have identified various genes and proteins involved in regenerating the liver. This includes the protein Shp2 which instructs liver cells to divide. However, scientists have found mice lacking the gene for Shp2 could still repair their livers. But how exactly these genetically modified mice were able to do this remained unclear.

To investigate, Kaneko et al. examined the shape and size of cells in the livers of mice lacking Shp2. This revealed clusters of dividing cells that could still repair the liver that contained abundant amounts of a protein called CD133. The CD133 molecules resided in very small vesicles about 50 to 150 nm in width which Kaneko et al. named 'intercellsomes' because they could move from one liver cell to the next.

Further experiments revealed that the intercellsomes contained important materials essential for cell division, making them distinct from other well-known vesicles. These newly discovered structures may allow liver cells to share replication signals with other cells that may be struggling to divide during liver regeneration.

CD133 is also present in cancer cells that are resistant to treatment and can multiply under stress. Kaneko et al. found that treating various types of tumor cells with drugs that inhibit proliferation led to an increase in CD133. This suggests that some cancer cells may use the intercellsome mechanism to keep dividing following treatment, potentially resulting in a relapse of the malignant disease.

Taken together, this study hints at the existence of a previously unknown communication system that helps cells to divide when their replication is inhibited. Further experiments are needed to see if this mechanism is widely employed by various cell types, how exactly the CD133 vesicles migrate between cells, and if intercellsomes carry out any other roles.

regarded as a source of progenitor cells that contribute to liver regeneration (*Dorrell et al., 2011*; *Miyajima et al., 2014*), although the identity and mechanism are yet to be deciphered.

Detected immediately following PHx was a rapid increase of growth factors in the liver and blood, including HGF, EGF, IL-6, and TNFα (*Michalopoulos, 2017*), accompanied by upregulation of proliferative signaling events, especially the Ras-Erk pathway, in hepatocytes. Shp2/Ptpn11, an SH2-containing tyrosine phosphatase, is a key molecule that promotes signaling from receptor tyrosine kinases (RTKs) to the Ras-Erk pathway (*Chan and Feng, 2007*; *Chen et al., 2016*; *Grossmann et al., 2010*; *Matozaki et al., 2009*; *Tajan et al., 2015*). Ablating Shp2 in hepatocytes attenuated hepatocyte proliferation following PHx and also suppressed RTK-driven hepato-oncogenesis (*Bard-Chapeau et al., 2006*; *Chen et al., 2021*; *Liu et al., 2018*). Of note, liver regeneration was impaired but not blocked by the removal of Shp2 or other pro-proliferative molecules in hepatocytes (*Fausto et al., 2006*; *Michalopoulos and DeFrances, 1997*; *Taub, 2004*). Thus, one critical question is why and how a small number of hepatocytes overcome the diminished mitogenic signaling, to re-gain proliferative capacity.

In this study, we found that in Shp2-deficient livers, actively dividing hepatocytes formed distinctive clusters and highly expressed CD133, but were negative for other stem/progenitor cell markers. The CD133+ cell colonies appeared transiently following PHx and disappeared quickly after the completion of liver regeneration. This phenotype was also observed in MET-deficient livers following liver injuries. CD133 expression was upregulated by proliferative signal inhibition in a variety of cancer cell lines from different organs, indicating a commonly shared mechanism. Remarkably, we identified and isolated CD133+ intracellular vesicles, which were enriched with transcripts of immediate early responsive genes (IEGs) and migrated between tightly contacting cells in colonies. Single-cell RNA-sequencing (scRNA-seq) demonstrated that cells defective for proliferative signaling exhibited lower intracellular transcriptional diversity (entropy), which was remedied in CD133+ cells, potentially by converting intercellular heterogeneity into intracellular diversity through sharing of mRNAs. These data suggest a new function of CD133 in intercellular

communication, by which normal and cancer cells endeavor to proliferate under proliferative signal deficit.

## Results

### Shp2 deficiency induces clustering of proliferating cells transiently in regenerating liver

We chose mouse liver regeneration to dissect mechanisms of cell proliferation in mammals under a physiological context. In previous experiments, we showed that the removal of Shp2 suppressed Ras-Erk signaling and hepatocyte division following PHx in mice (*Bard-Chapeau et al., 2006*). However, it was unclear why certain Shp2-negative hepatocytes managed to divide when proliferative signaling was impaired in general. To address this question, we examined carefully the regenerating process in hepatocyte-specific Shp2 knockout (SKO, *Shp2flox/flox:Alb-Cre*) livers. At 2 days after PHx, WT and SKO livers exhibited similar overall and histological phenotypes (*Figure 1—figure supplement 1A and B*). Immunostaining for Ki67 showed significantly lower numbers of proliferating hepatocytes in SKO than WT livers, with no difference for non-parenchymal cells (NPCs) where Shp2 was not deleted (*Figure 1A*, *Figure 1—figure supplement 1C, and 1D*). Intriguingly, we identified certain areas that were enriched with proliferating hepatocytes in SKO liver (*Figure 1A*; arrow). This peculiar proliferation pattern was especially cryptic, because these areas did not display severe inflammation or different phenotypes of hepatocytes except active proliferation (*Figure 1A*; right panel). Therefore, we searched extensively for biomarker molecules, which could distinguish the actively proliferating cells in the particular areas, including GFAP, EpCAM, CK19, CD133, Sox9, CD44, and α-fetoprotein (AFP) (*Figure 1—figure supplement 1E–1H*). Of note, CD133 was the only molecule that was highly expressed in the areas enriched with proliferating hepatocytes (*Figure 1B*). The percentages of Ki67+ hepatocytes were significantly higher in CD133+ than in CD133− areas in regenerating livers (*Figure 1C and D*). These patchy areas appeared 2 days after PHx, and underwent expansion until 3 weeks, whereas WT livers did not show comparable CD133 expression in hepatocytes (*Figure 1E*). Liver/body weight ratios were similar between SKO and WT mice by 3 weeks (*Figure 1—figure supplement 1I*), suggesting that the active division of CD133+ cells could compensate for the impaired proliferative capacity of CD133− hepatocytes in SKO liver.

The CD133+ hepatocytes also expressed high levels of E-cadherin and displayed tight cell-cell contact in the colonies (*Figure 2A*, *Figure 1—figure supplement 1J and K*). CD133 was localized to the apical side of hepatocytes (*Figure 2A*), and was rarely detected in other types of cells located on their basal sides. At 3 weeks after PHx, the colonies appeared more distinctive even in H&E staining, which contained smaller hepatocytes with compact structure, albeit no difference observed for NPC types and the vasculature (*Figure 2B*). Immunostaining for sinusoidal endothelial cells and HGF did not show any unique features in the colonies otherwise (*Figure 1—figure supplement 1L and M*). The distinct colonies merged into surrounding tissues 5 weeks after PHx, as evidenced by loss of CD133 expression and disappearance of clear colony edges (*Figure 2C*). Therefore, the CD133+ colonies of proliferating hepatocytes were composed transiently during liver regeneration.

Due to the compacting colony architecture 3 weeks after PHx, we observed a clear difference in light scattering/refraction in unstained thick liver sections (*Figure 2D*), which allowed manual dissection of the colonies out of liver sections for analysis. Immunoblotting confirmed the enrichment of CD133+ and E-cadherin+ cells but barely detected Shp2 protein in the isolated colonies, indicating their origin from Shp2-deleted hepatocytes (*Figure 2D and E*). CD133 is known to be expressed in oval cells, which can differentiate into both hepatocytes and bile duct epithelial cells, similar to the potential of hepatoblasts (*Figure 2F*). However, we did not observe the expression of other oval cell markers, including EpCAM, AFP, and Sox9, in the colony lysates, (*Figure 2E*), consistent with the immunostaining data (*Figure 1—figure supplement 1E–1H*). Indeed, hepatocytes in the colonies did not exhibit oval cell-like morphology throughout the regenerative process (*Figures 1A, E and 2A–C*). CD133 expression in colonized hepatocytes was even higher than the embryonic liver at E17.5, which highly expressed AFP, a hepatoblast marker (*Figure 2E*). The molecular size of CD133 expressed in the colonies was slightly different from that in the bile duct (*Figure 2E*), likely reflecting variations in glycosylation (*Barkeer et al., 2018*; *Kemper et al., 2010*). Furthermore, the colony-forming cells expressed a mature hepatocyte marker HNF4α, at similar levels to non-colony areas (*Figure 2E*).

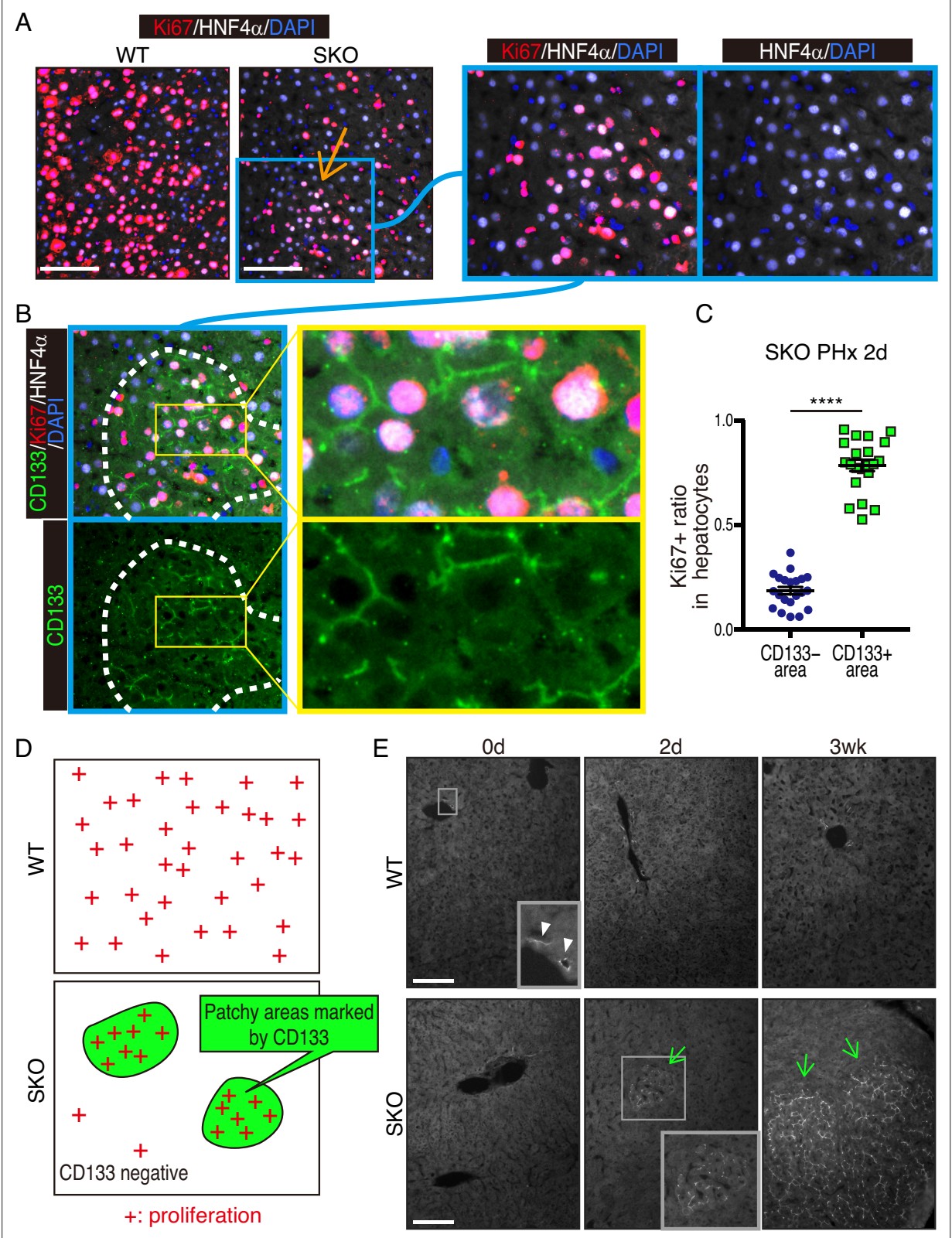

**Figure 1.** Identification of a patchy hepatocyte proliferation pattern in Shp2-deficient (SKO) liver after partial hepatectomy (PHx). (**A and B**) Immunofluorescence on liver tissue sections 2 days after PHx. HNF4α is a hepatocyte marker and Ki67 is a proliferation marker. Arrow in (**A**) points to an area enriched with proliferating hepatocytes. Dashed line in (**B**) shows an area with continuous CD133 expression. (**C**) Quantification of proliferating rate in hepatocytes in CD133-positive and -negative areas in Shp2 knockout (SKO) livers 2 days after PHx. Each dot indicates one area. Data were

*Figure 1 continued on next page*

*Figure 1 continued*

collected from 3 mice. Means ± SEM are shown. ****p<0.0001 (two-tailed unpaired t-test). (**D**) While WT hepatocytes proliferated at high frequency everywhere, proliferating hepatocytes in SKO liver were mostly located in patchy areas marked by CD133 expression. (**E**) Immunofluorescence of CD133 on liver tissues at day 0 or 2, and 3 weeks (0d, 2d, 3 wk) after PHx. CD133⁺ hepatocyte clusters were only found in SKO livers after PHx (light green arrows). In WT livers, CD133 expression was only seen in bile duct epithelial cells (arrowheads). Scale bars, 100 μm (**A and E**).

The online version of this article includes the following figure supplement(s) for figure 1:

**Figure supplement 1.** Liver regeneration of WT and Shp2 knockout (SKO) livers after partial hepatectomy (PHx).

These results indicate that the clustered CD133⁺ cells in SKO liver are not hepatoblasts or oval cells but are mature hepatocytes that acquire unique proliferation potential following PHx.

## The proliferative cell colonies originate from heterogeneous mature hepatocytes

To determine the origin of these colonized hepatocytes by clonal tracing, we transfected vectors carrying a GFP and a *sleeping beauty* (SB) transposon (*Ivics et al., 1997*) into the liver by hydrodynamic tail vein injection (HTVi) (*Liu et al., 1999*; *Zhang et al., 1999*), which labeled a few hepatocytes with GFP in a mosaic pattern (*Figure 3A, B*; *Mikkelsen et al., 2003*; *Yant et al., 2000*). We then performed PHx and analyzed regenerating livers 3 weeks later (*Figure 3A, B*). The resulting colonies in SKO livers contained both GFP⁺ and GFP⁻ cells (*Figure 3B*), indicating multiple origins of hepatocytes. When a large GFP⁺ cell clone was located at a colony edge, we did not see the recruitment of GFP⁻ cells, even though the surrounding hepatocytes were mostly GFP-negative (*Figure 3C*). Thus, the colonies were expanded by the proliferation of cells from inside rather than by continuous recruitment of surrounding hepatocytes (*Figure 3—figure supplement 1A*). By measuring clone sizes of GFP⁺ hepatocytes inside and outside, we found that hepatocytes within the colonies divided much more actively than those located outside (*Figure 3D, E*), suggesting that clustering enabled hepatocytes to divide actively (*Figure 3F*). The starting colony size at the fate determination stage (*Figure 3F*) was estimated to be approximately 10 hepatocytes, based on the GFP⁺ clone/colony size ratios (*Figure 3G* and *Figure 3—figure supplement 1B*). The initial colony size should be large enough to include a variety of hepatocytes, signifying that the colony-forming potential does not require a special property of hepatocytes. The existence of mononuclear and binuclear cells (*Duncan et al., 2010*) in the colonies 2 days after PHx further supports their heterogeneity in origin (*Figure 3—figure supplement 1C*).

We reintroduced WT Shp2 together with GFP into a few Shp2-deficient hepatocytes by HTVi, and analyzed the mosaic liver tissues 2 days after PHx (*Figure 3—figure supplement 2A*). Re-introducing Shp2 indeed restored the WT hepatocyte proliferation pattern in a cell-autonomous manner (*Figure 3—figure supplement 2B, C*). Although the colonies were mainly composed of Shp2-negative hepatocytes (*Figure 3—figure supplement 2D*), we observed Shp2⁺ cells inside the colonies, which were also CD133⁺ (*Figure 3—figure supplement 2E*). Thus, Shp2 deficiency was not a requirement to join the colony, although Shp2⁺ cells disturbed colony morphology (*Figure 3—figure supplement 2E*). Shp2-negative hepatocytes likely sent out paracrine signals to trigger clustering with surrounding hepatocytes, including Shp2⁺ cells in the neighborhood.

A similar pattern of colony formation was observed in cultures of primary hepatocytes isolated from SKO liver. Shp2-negative hepatocytes formed unique colonies of Ki67⁺ cells in vitro, resembling the observed clusters in liver sections (*Figure 3H* and *Figure 3—figure supplement 1D*), while proliferating hepatocytes were distributed rather randomly in the WT culture (*Figure 3—figure supplement 1D*). E-cadherin expression in cell-cell contacts distinguished these colonies (*Figure 3H*), featuring their similar properties to those in vivo. This phenotype was independent of colony sizes and, therefore, was not due to high cell density in the area (*Figure 3H*). The colony formation in primary hepatocyte culture also confirms their origin from mature hepatocytes.

## Cluster formation of dividing cells is a common mechanism under stresses

CCl₄ is metabolized by hepatocytes around the central veins into a hyper-oxidative chemical that causes necrosis and apoptosis in the area (*Forbes and Newsome, 2016*; *Weber et al., 2003*). We

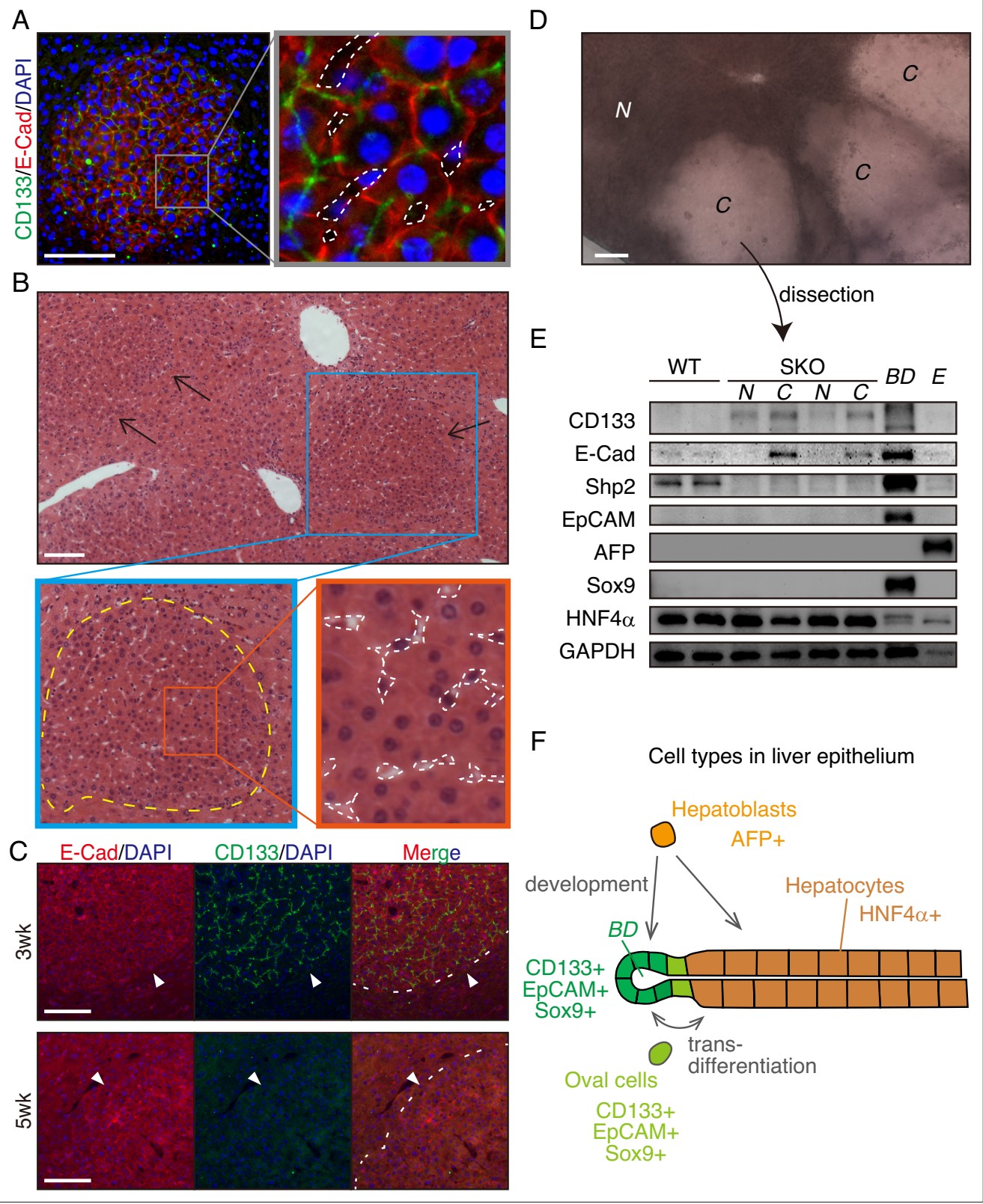

**Figure 2.** CD133+ colonies represent a unique regeneration process in Shp2-deficient liver. (**A**) Representative image of CD133+ colonies shown by immunofluorescence on SKO liver tissues 3 weeks after partial hepatectomy (PHx). Dashed lines, vasculatures. (**B**) H&E staining of Shp2 knockout (SKO) liver sections 3 weeks after PHx. Yellow dashed line: boundary between the colony and surrounding tissue; white dashed lines: vasculatures. (**C**) Immunofluorescence on SKO liver sections at indicated time points after PHx. Arrowheads and dashed lines indicate the boundaries between the

*Figure 2 continued on next page*

*Figure 2 continued*

colonies and surrounding tissues, which were clear at 3 weeks but disappeared at 5 weeks. (**D**) Unstained thick tissue section of SKO liver 3 weeks after PHx. (**E**) Colonies (**C**) and non-colony control areas (**N**) were dissected out from thick tissue sections of SKO livers 3 weeks after PHx, and analyzed by immunoblotting. Random areas from WT livers 3 weeks after PHx, bile duct (*BD*), and E17.5 liver (*E*) were used for comparison. (**F**) Illustration of epithelial lineages in the liver. Scale bars, 100 µm (**A–D**).

The online version of this article includes the following source data for figure 2:

**Source data 1.** Source data for western blot in panel E.

examined CD133 induction in the liver following $CCl_4$-induced injury, which involves cell death and inflammation, largely different microenvironment from the PHx model. At 2 days after the $CCl_4$ injection, we detected distinctive $CD133^+$ hepatocyte clusters in SKO livers with a higher proliferation rate than the surrounding areas (*Figure 4A*). Although $CCl_4$ also induced ductular reaction by $CD133^+$ bile duct epithelial cells (*Miyajima et al., 2014*), these areas were not associated with the $CD133^+$ hepatocyte clusters (*Figure 4A, B*; arrowheads). Consistent with the PHx model, while bile duct epithelial cells were $EpCAM^+$, $CD133^+$ hepatocytes did not express EpCAM (*Figure 4A*). The $CD133^+$ hepatocyte colonies appeared around the injured area, consistent with the notion of midlobular hepatocytes located next to a pericentral injury site (*Wei et al., 2021*).

We further examined the effect of hepatocyte-specific deletion of HGF receptor MET in mice. Similar $CD133^+$ proliferating hepatocyte clusters were induced by PHx or $CCl_4$ injection in MET-deficient livers (*Figure 4C*). These results suggest that CD133 induction is not restricted to a specific gene deletion or an injury model, but rather is a response to impaired proliferative signaling. Indeed, treatment of PLC liver cancer cells in vitro with a Shp2 inhibitor (SHP099) or Mek inhibitor (Trametinib), or a Shp2-targeting CRISPR vector, also upregulated CD133 expression (*Figure 4D, E*), indicating that CD133 induction is an inherent cellular response to impaired signaling of the RTK-Ras-Erk pathway. Upregulated CD133 expression was independent of other stem cell markers and thus did not reflect the forced selection of a stem cell population (*Figure 4D*). Moreover, CD133 induction was rapid and dramatic in a manner that cannot be explained by selective expansion of pre-existing $CD133^+$ cells.

To determine if it is a widely conserved mechanism, we examined CD133 induction in a variety of cell lines derived from various organs and animals. Treatment with Shp2 or Mek inhibitor upregulated CD133 expression in mammary epithelial cells MCF10A (*Figure 4F, G*), glioblastoma cells GSC3028 and GSC23, insulinoma cells MIN6, colon adenocarcinoma cells MC38, and cervical cancer cells Hela as well as kidney fibroblasts COS7 (*Figure 4H*). Together, these results suggest that CD133 upregulation under proliferative signal deficit is a common mechanism in various cell types.

## Upregulated CD133 is located on filament fibers connecting neighboring cells

Since the $CD133^+$ cell clustering is apparently triggered by a local intercellular signal, we examined the expression of molecules known to mediate intercellular communication at a short distance (*Lander, 2007*; *Mikels and Nusse, 2006*; *Nusse, 2003*; *Rogers and Schier, 2011*; *Teleman et al., 2001*; *Tsiairis and Aulehla, 2016*). We detected significantly higher expression of Wnt7a, Wnt10a, and Shh in SKO liver (*Figure 3—figure supplement 2F*). Consistently, porcupine (PORCN), a protein that resides in the ER and modifies Wnt for secretion (*Clevers et al., 2014*; *Mikels and Nusse, 2006*; *Nusse, 2003*; *Tammela et al., 2017*), was highly expressed in clustered hepatocytes (*Figure 3—figure supplement 2G*), suggesting a role of local Wnt signaling in colony assembly. PORCN expression was also strongly associated with the colonies in cultured primary SKO hepatocytes in vitro (*Figure 3—figure supplement 2H*), and PORCN protein amounts were higher in SKO than WT liver lysates 2 days after PHx (*Figure 3—figure supplement 2I*). Furthermore, the mRNA level of Wnt10a was drastically elevated in response to PHx in SKO liver (*Figure 3—figure supplement 2J*). To probe the cellular resource, we isolated hepatocytes and NPCs from SKO livers 2 days after PHx. Wnt10a was mainly produced by hepatocytes, whereas the major source of Shh might not be hepatocytes (*Figure 3—figure supplement 2K*). These results combined with the Shp2 rescue experiment suggest that the Wnt10a paracrine signal from SKO hepatocytes might be a local inducer of the colonies. Treatment of human HCC cells with Trametinib upregulated Wnt10a expression, and additional inhibition of PORCN suppressed

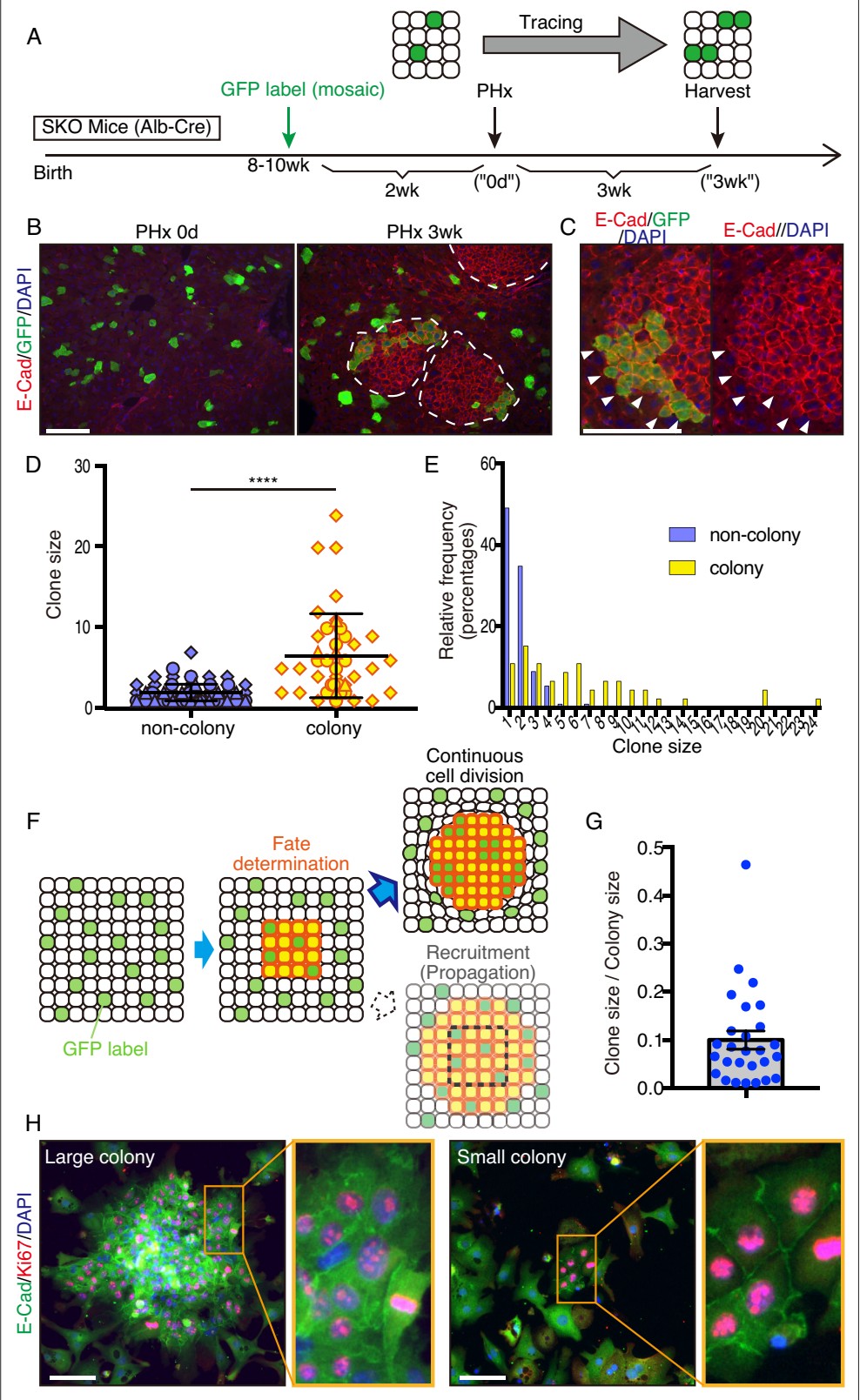

**Figure 3.** Mitogenic signal deficiency induced tight clustering of cells that leads to continuous growth. (**A**) Schematic illustration of clonal tracing performed in Shp2 knockout (SKO) livers during regeneration. (**B**) Liver tissues were examined before (PHx 0d) or 3 weeks after the surgery (PHx 3 wk). E-Cad shows the colony structures (arrows). (**C**) GFP-labeled clone in a colony 3 weeks after PHx. Note that no unlabeled cells were detected on this

*Figure 3 continued on next page*

*Figure 3 continued*

edge (arrowheads). (**D**) Cell numbers of GFP-labeled hepatocyte clones in colony and non-colony areas, counted on sections. Data were collected 3 weeks after PHx. Means ± SD are shown, n=3, ****p<0.0001 (Mann-Whitney test). (**E**) Distribution analysis of clone sizes from (**D**). (**F**) Clonal dynamics of colony-forming hepatocytes. (**G**) Clone/colony size ratios to estimate the original colony size (see also *Figure 3—figure supplement 1B*). Mean ± SEM is shown. (**H**) Immunofluorescence on the colonies (E-Cad⁺ clusters) in Shp2-deficient hepatocyte culture in vitro. Note the other non-colony cells with low proliferation rate, indicating the patchy proliferation in the colonies. See also *Figure 3—figure supplement 1D*. Scale bars, 100 µm (**B, C, and H**).

The online version of this article includes the following source data and figure supplement(s) for figure 3:

**Figure supplement 1.** Clonal analysis of hepatocytes.

**Figure supplement 2.** Molecular analysis of intercellular signals during colony induction.

**Figure supplement 2—source data 1.** Source data for western blot in panel I.

CD133 induction (*Figure 3—figure supplement 2L*), suggesting a role of Wnt10a in CD133 induction under proliferative signal deficiency.

Interestingly, we observed CD133 on filament-like structures in colonized E-cadherin⁺ cells in the primary SKO hepatocyte culture (*Figure 5A*). These CD133⁺ filament-like structures bridged neighboring cells in the colonies (*Figure 5A, B*), suggesting a direct cell-cell communication event. This direct communication model is consistent with the distinctive feature of clustered cells with extremely tight contacts in vivo and in vitro (*Figures 1B, D, Figure 2A, Figure 3F and H*). Given the filament-like staining pattern, we examined PLC cells with an antibody against human CD133. Indeed, we observed similar CD133⁺ filament structures inside and also connecting the cells (*Figure 5C*). Of note, the antibodies used for the mouse and human cells are different and do not cross-react with the other species. To further rule out a possible artifact in immunostaining for CD133, we expressed a Myc-tagged CD133 fusion protein in PLC cells and stained with antibodies against CD133 and the Myc-tag. The two signals overlapped on the filament-like structure, confirming the subcellular localization of CD133 (*Figure 5—figure supplement 1A, B*). As CD133 possesses five transmembrane domains and a membrane-targeting signal peptide, we postulated that the observed filament-like CD133 staining represents CD133⁺ vesicles on filaments. Indeed, the CD133 signal co-localized with tubulin filaments (*Figure 5—figure supplement 1C*). CD133 overexpression led to a thicker pattern of CD133⁺ filaments, especially at high expression levels (*Figure 5—figure supplement 1A - 1C*). Similar CD133⁺ structure was induced by Shp2 and MET inhibitors in MCF10A, HeLa, and MC38 cells (*Figure 5D* and *Figure 5—figure supplement 1D*). Thus, these data suggest that CD133 is located on a trafficking machinery in proliferating cells.

## CD133⁺ vesicles are induced by defective proliferative signaling

The filamentous staining profile of CD133 was different from the conventional droplet-like patterns often observed with endosomes or exosomes in a multi-vesicular body. To determine the subcellular localization of CD133 and the CD133⁺ filaments, we performed stochastic optical reconstruction microscopy (STORM) imaging. Indeed, the CD133⁺ filaments observed at low resolution were CD133⁺ vesicles tightly associated with and distributed on tubulin filaments at super-resolution (*Figure 5E*). CD133 overexpression modified the morphology and topology of the vesicles (*Figure 5F*), suggesting a role of CD133 in the vesicle assembly and dynamics. Quantitative co-localization analysis confirmed the physical location of CD133⁺ vesicles on tubulin filaments, while the value decreased after CD133 overexpression (*Figure 5E, F*, and *Figure 5—figure supplement 1E*). The lowered co-localization value following CD133 overexpression likely reflected the altered morphology of the vesicles, stretching away from the tubulin filaments, which also explained the bulky appearance in low-resolution images (*Figure 5—figure supplement 1A–1C*). While this function is consistent with a reported role of CD133 on lipid membrane morphology/topology (*Röper et al., 2000*; *Thamm et al., 2019*), we observed CD133 on cell surface membrane only when exogenous CD133 was overexpressed (*Figure 5—figure supplement 1B and F*). In agreement with STORM imaging, immunofluorescence, and immunogold electron microscopy on a cryo-ultramicrotome section of the colonies in regenerating SKO liver further confirmed the localization of CD133 on intracellular vesicles enriched between the apical lumens, rather than distributed on the apical membrane surface (*Figure 5G* and *Figure 5—figure supplement 1G*).

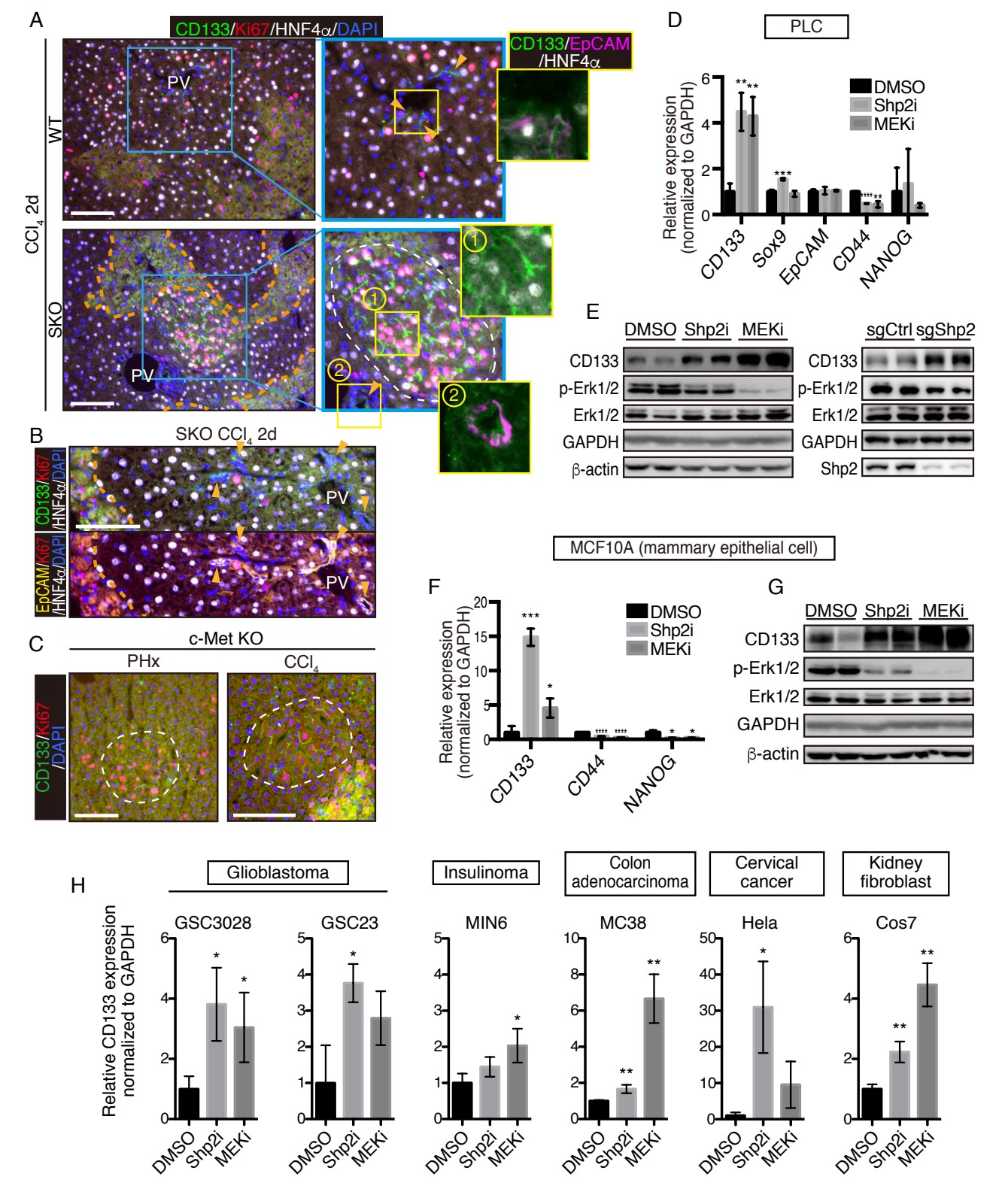

**Figure 4.** CD133 induction during signal deficiency is a widely conserved mechanism. (**A and B**) Immunofluorescence of liver sections 2 days after CCl₄ injection. Proliferating hepatocytes were scattered in WT livers, whereas they were highly concentrated in CD133⁺ colonies (white dashed line) in Shp2 knockout (SKO) livers as shown in (**A**). CD133⁺/EpCAM⁺/HNF4α⁻ bile duct epithelial cells (arrowheads) were not associated with the colonies. Pink dashed lines, injured areas. PV, portal vein. (**C**) Immunofluorescence of Met[hep-/-] liver sections 2 days after partial hepatectomy (PHx) or CCl4 injection.

*Figure 4 continued on next page*

*Figure 4 continued*

White dashed lines, CD133⁺ colonies. Pink dashed line, injured area. (**D**) qRT-PCR analysis of PLC cell lysates treated with Shp2 or MEK inhibitors (Shp2i and MEKi). \*\*p<0.01, \*\*\*p<0.001, \*\*\*\*p<0.0001 (two-tailed unpaired t-test, each compared with DMSO treatment). Means ± SD from three replicates are shown. (**E**) Immunoblotting of PLC cell lysates treated with inhibitors or transfected with Shp2 targeting CRISPR vector. Guide RNA targeting the AAVS1 safe harbor site was used as a control (sgCtrl). (**F**) qRT-PCR analysis of MCF10A cell lysates treated with the inhibitors. \*p<0.05, \*\*\*p<0.001, \*\*\*\*p<0.0001 (two-tailed unpaired t-test, each compared with DMSO treatment). Means ± SD from three replicates are shown. (**G**) Immunoblotting of MCF10A cell lysates treated with inhibitors. (**H**) qRT-PCR analysis of lysates from various cell lines treated with Shp2 or MEK inhibitors. \*p<0.05, \*\*\*p<0.001, \*\*\*\*p<0.0001 (two-tailed unpaired t-test, each compared with DMSO treatment). Means ± SD from three replicates are shown. Scale bars, 100 µm (**A–C**).

The online version of this article includes the following source data for figure 4:

**Source data 1.** Source data for western blots in panel E and G.

---

We isolated CD133⁺ vesicles from PLC cells using magnetic beads bound with anti-CD133 antibody, with the yields dramatically increased after pre-treatment with Trametinib (*Figure 5H*). Visualized under EM, the sizes of CD133⁺ vesicles were estimated around 50 nm (*Figure 5I*), relative to EVs/exosomes that are usually at the sizes of 30–200 nm. Notably, CD133⁺ vesicles did not contain exosome markers CD9, CD63, or CD81, supporting their identity as a new type of vesicle, different from the classical exosomes (*Figure 5H*). Co-staining endosome markers, RAB5A and RAB7, showed no overlap with CD133, with no CD133 signal detected in the multivesicular body where exosomes accumulated (*Figure 6—figure supplement 1A*). Evidently, CD133 was located on a filament network that connects neighboring cells, instead of endosomes or exosomes. Therefore, we give the name 'intercellsome' to this new type of vesicle identified here.

## CD133⁺ intercellsomes contain unique cargos

We isolated CD133⁺ vesicles from SKO liver lysates following PHx, and found that the vesicles were not enriched with signaling molecules, such as Stat3 and Erk1/2, compared to the CD133⁻ fractions (*Figure 6—figure supplement 1B*), suggesting that proliferative signaling proteins were not likely the most critical cargos. Interestingly, the CD133⁺ vesicles were markedly different from CD133⁻ fractions in the overall RNA profiling and contained a minimal amount of rRNAs or micro-RNAs, with the majority being mRNAs (*Figure 6A*). The expression of immediate early-responsive genes (IEGs), including *Jun, Junb, and Myc,* were upregulated rapidly following PHx (*Fausto, 2000*; *Haber et al., 1993*). qRT-PCR analysis showed that these mitogenic mRNAs were loaded in CD133⁺ vesicles, with no enrichment detected for housekeeping gene transcripts, such as *Gapdh* and *Ppia*, (*Figure 6B* and *Figure 6—figure supplement 1C*), showing selective inclusion of mRNAs. Similar results were obtained with CD133⁺ vesicles isolated from PLC cells treated with Trametinib (*Figure 6—figure supplement 1D*).

We performed RNA-seq analysis of CD133⁺ vesicles isolated from Trametinib-treated PLC cells, extracellular vesicles (EVs) secreted by the same cells, and the whole cells. Isolation of the EV fraction was performed by the polymer-based precipitation method from the culture supernatant. Small RNAs, including miRNAs, snoRNAs, and snRNAs, were less abundant in CD133⁺ vesicles than the whole cells (*Figure 6C*). In contrast, miRNAs were enriched in the EVs compared to the whole cells (*Figure 6C*). Although the majority of RNA species detected in EVs were still mRNAs, the data could be influenced by the library preparation with total RNAs, which missed lots of small RNAs. Nevertheless, the enrichment of miRNAs in the EVs was in sharp contrast to significantly enriched mRNAs in CD133⁺ vesicles (*Figure 6C*), clearly distinguishing the two types of vesicles. Compared to the whole cells, 4309 mRNAs were upregulated, with 3974 mRNAs downregulated, in CD133⁺ vesicles (*Figure 6C*). In particular, the RNA-seq detected enrichment of IEG transcripts in CD133⁺ vesicles, while most of these mRNAs were at low levels in the EVs (*Figure 6D* and *Figure 6—figure supplement 1E*). Consistently, single molecule in situ hybridization showed localization of *MYC* mRNA on CD133⁺ filaments, verifying the IEG existence in these vesicles (*Figure 6E, F*).

To trace CD133⁺ vesicle activity, we established a GFP-labeled cell line that also expressed stably a CD133-Myc fusion protein. After mixing these cells with mCherry-expressing cells without the fusion protein, we traced the Myc-tag signal (*Figure 6G*). Indeed, the CD133-Myc protein was transported from GFP⁺ cells to the neighboring mCherry⁺ cells (*Figure 6H*). Importantly, neither GFP nor

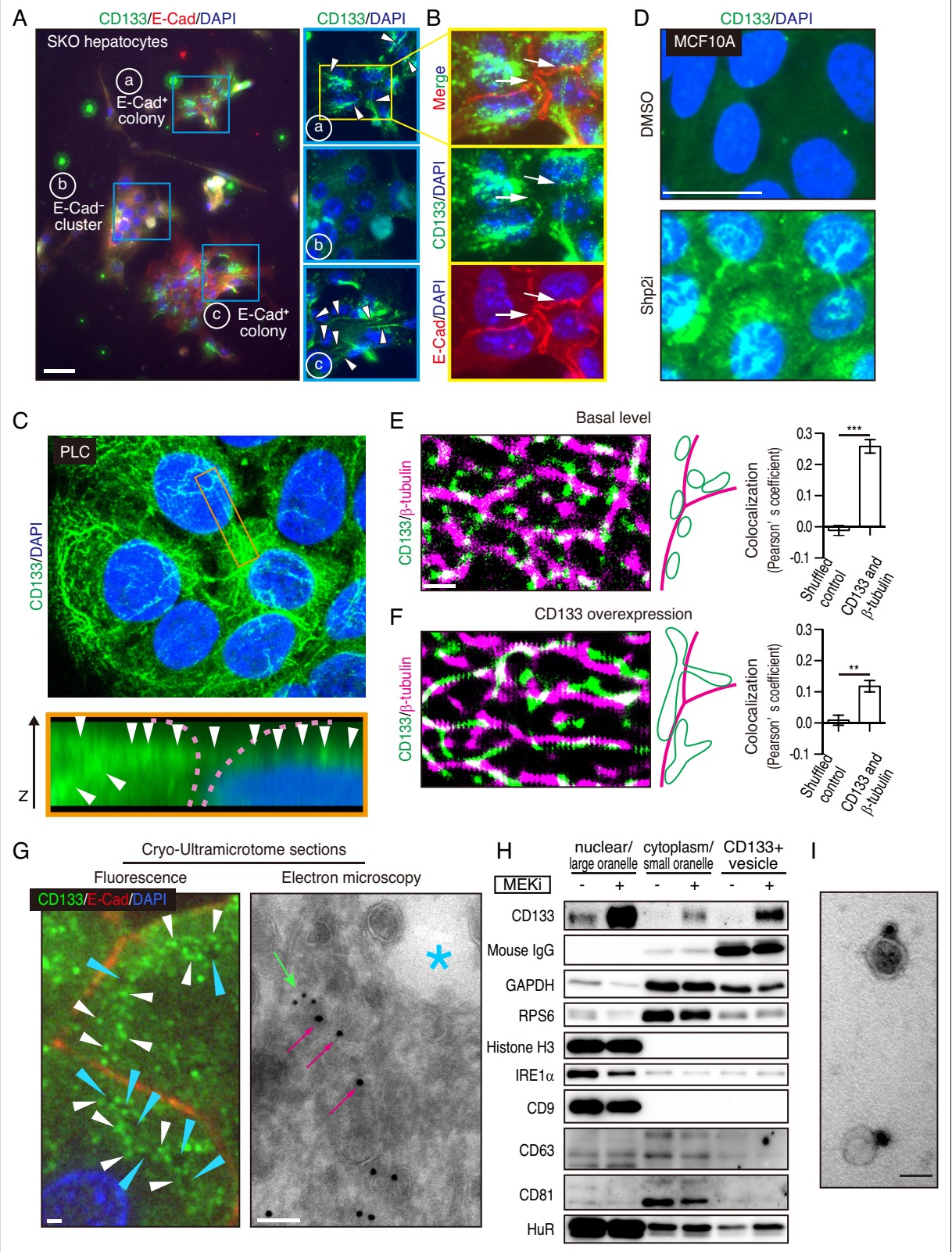

**Figure 5.** Mitogenic signal deficiency induces CD133⁺ vesicles. (**A and B**) Immunofluorescence on in vitro colonies of primary hepatocytes from Shp2 knockout (SKO) liver. CD133 was localized at filament-like structures in E-Cad⁺ colonies as shown by arrowheads in (**A**), which were connected between different hepatocytes as shown by arrows in (**B**). (**C**) 3D-reconstituted confocal image of immunofluorescence on PLC cells. Lower panel shows the Z-plane section of the orange box area. Arrowheads indicate the CD133 signal on continuous filament-like structures bridged between neighboring

*Figure 5 continued on next page*

*Figure 5 continued*

cells. Pink dashed lines indicate the cell surface. (**D**) Immunofluorescence of MCF10A cells treated with Shp2 inhibitor. (**E and F**) Super-resolution STORM images of immunofluorescence on PLC cells without (**E**) or with (**F**) CD133 overexpression. Colocalization of CD133 and β-tubulin was analyzed as Pearson's coefficient. Mismatched green and magenta channels from shuffled ROIs were measured as controls (Figure S4E). Means ± SEM from six images are shown. \*\*p<0.01, \*\*\*p<0.001 (two-tailed unpaired t-test). (**G**) Immunofluorescence and Immuno-Gold EM images of cryo-ultramicrotome sections of SKO liver tissue after partial hepatectomy (PHx). Cyan arrowheads and asterisk indicate apical lumens. White arrowheads indicate the CD133 signals aligned between the apical lumens of neighboring cells. Light green arrow, CD133 staining (12 nm colloidal gold); Magenta arrows, α-tubulin staining (18 nm colloidal gold). (**H**) Immunoblotting of CD133+ vesicles isolated from MEK inhibitor (MEKi) -treated PLC cells. Markers for different fractions were analyzed. CD133 antibody used for the vesicle isolation was IgG produced in mouse, which was detected by anti-mouse IgG antibody, showing efficient capture by the beads. Despite the efficient capture, the DMSO-treated PLC cells did not have much CD133+ vesicle to be bound with the antibody. (**I**) EM image of Immunogold staining on the isolated vesicle. Scale bars, 100 µm (**A**), 25 µm (**D**), 1 µm (E, F, Fluorescence in G), 100 nm (EM in G), 50 nm (**I**).

The online version of this article includes the following source data and figure supplement(s) for figure 5:

**Source data 1.** Source data for western blot in panel H.

**Figure supplement 1.** Analyses of CD133 localization.

mCherry signals migrated between the cells, suggesting selective CD133+ vesicle traffic intercellularly. We further established a cell line that stably expressed a CD133-GFP fusion protein and mixed it with mCherry-expressing cells without the fusion protein. Correlative light and electron microscopy (CLEM) confirmed the traffic of CD133-GFP to the mCherry+ cells (*Figure 6—figure supplement 2*). Together, these results suggest a unique property of the vesicles, which contain particular cargos that are different from those in EVs.

## mRNA sharing converts intercellular heterogeneity into intracellular diversity of IEGs

HuR is an RNA-binding protein that binds to AU-rich elements in mRNAs for their stabilization, and shuttles between the nucleus and cytoplasm. This protein was loaded in CD133+ vesicles, as revealed by immunoblotting (*Figures 5H and 7A*) and was localized to the apical side of CD133+ hepatocytes (*Figure 7—figure supplement 1A*). Besides the nuclei, immunostaining detected HuR in some peri-nuclear areas (*Figure 7B*; arrows) and co-localization with CD133 in PLC cells (*Figure 7B*). The CD133+ filaments bridged neighboring cells with HuR visible on the bridging filaments (*Figure 7B*; arrowheads), suggesting migration of HuR-bound mRNAs directly from one cell to another. Treatment of PLC or MCF10a cells with Shp2 or Mek inhibitor markedly increased the density of HuR on CD133+ filaments (*Figure 7C* and *Figure 7—figure supplement 1B*), with even brighter HuR signals than the nuclei. Given that CD133 was found to modify the vesicle patterns (*Figure 5E, F*, *Figure 5—figure supplement 1A–1C*), we tested if CD133 overexpression also affected HuR trafficking. Indeed, the pattern of HuR changed together with altered CD133+ signals (*Figure 7D*). Lipofection of PLC cells with isolated CD133+ vesicles enhanced cyclin D1 expression, supporting a notion that mitogenic molecules were carried in the vesicles (*Figure 7E, F*, *Figure 6—figure supplement 1D*, *Figure 7—figure supplement 1C*). Colocalization of CD133 and HuR on the isolated vesicles was also demonstrated with an electron microscopy analysis (*Figure 7—figure supplement 1D*).

We wondered how a potential exchange of mitogenic mRNAs within cell colonies could boost proliferative capacity. If the cells with impaired signaling had equally reduced amounts and types of mitogenic mRNAs, an exchange would not be beneficial. We hypothesized that the expression of different IEGs was variably downregulated, which triggered the intercellular exchange of mRNAs through CD133+ intercellsomes (*Figure 7G*). This process may convert intercellular heterogeneity to higher intracellular IEG diversity (calculated as entropy), without increasing total IEG expression levels. To test this theory, we performed scRNA-seq of WT and SKO hepatocytes at 4 hrs and 2 days following PHx, using our newly established protocol (*Chen et al., 2021*; *Liang et al., 2022*). Hepatocyte identity was confirmed by unsupervised clustering and analyses of multiple markers, with contaminated NPCs excluded from further analyses (*Figure 7—figure supplement 2*). We focused attention on IEGs, which were highly induced following PHx (Methods) and were detected in isolated CD133+ vesicles. Indeed, the heatmap demonstrated stochastic expression profiles of different IEG species in individual cells (*Figure 7—figure supplement 3A*). CD133+ cells were easily identified in SKO liver due to its high expression (*Figure 7—figure supplement 3B*). Of note, CD133+ cells were not enriched with

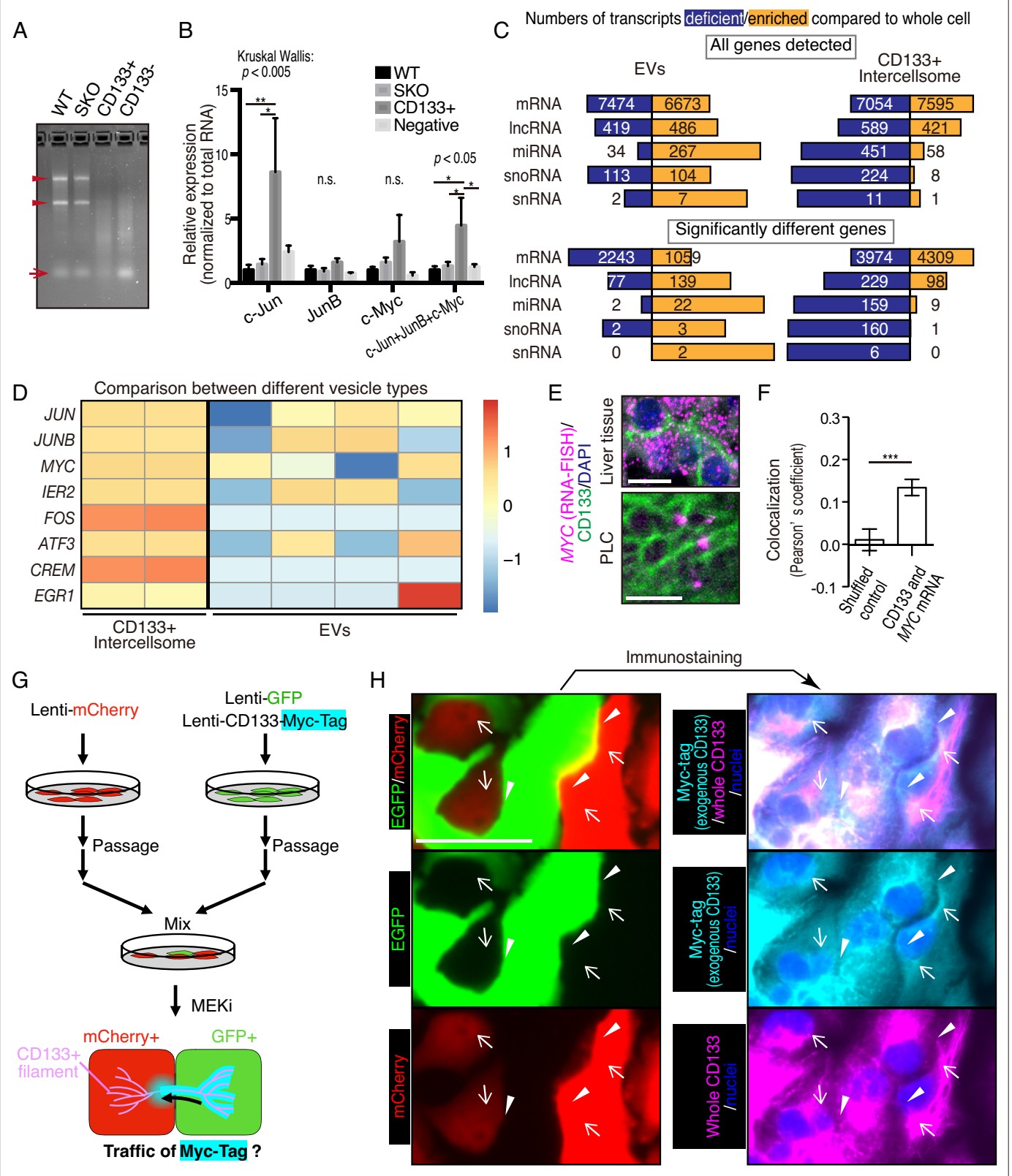

**Figure 6.** CD133+ vesicles contain mitogenic mRNAs and traffics between neighbor cells. (**A**) Agarose gel electrophoresis of total RNAs extracted from WT and Shp2 knockout (SKO) livers and CD133-positive vesicle and negative fractions from SKO liver after partial hepatectomy (PHx). Arrowheads show rRNAs and arrow shows microRNAs. (**B**) qRT-PCR analysis of RNAs extracted form WT (3 mice) and SKO tissues (3 mice) and from CD133+ vesicles and CD133− fractions (four mice for both). Means ± SEM are shown. *p<0.05, **p<0.01 (uncorrected Dunn's multiple comparison test, performed after

*Figure 6 continued on next page*

*Figure 6 continued*

Kruskal-Wallis test). n.s., not significant. (**C**) RNA-seq analysis of the different cell types and the whole cells. The bars indicate proportions between numbers of deficient and enriched gene transcripts in each RNA types. (**D**) Comparison of IEG contents between the different vesicle types with RNA-seq. Different lanes indicate independent vesicle isolations. (**E**) RNA-FISH for *Myc* mRNA and immunostaining for CD133. (**F**) Quantitative colocalization analysis of CD133 and MYC mRNA in PLC cells shown in (**E**). Means ± SEM from 14 images are shown. (**G**) Experimental design to detect the traffic of CD133⁺ vesicles between neighbor cells. (**H**) Immunostaining of Myc-tag and CD133 on GFP⁺ and mCherry⁺ PLC cells mixed as shown in (**G**). Note that Myc-tag only indicates exogenous CD133, while CD133 indicates both endogenous and exogenous CD133. Myc-tag was primarily detected in the GFP⁺ cells, but also detected on the bridges (arrowheads) and in the mCherry⁺ cells (arrows). GFP was not detected at the same locations (arrowheads and arrows), indicating the specific traffic of CD133-Myc-tag. Scale bars, 10 µm (**E**) and 50 µm (**H**).

The online version of this article includes the following source data and figure supplement(s) for figure 6:

**Figure supplement 1.** Analyses of CD133⁺ vesicles.

**Figure supplement 1—source data 1.** Source data for western blot in panel B.

**Figure supplement 2.** Intercellular traffic of CD133⁺ vesicles.

particular hepatocyte subtype markers or stem cell markers, compared to CD133⁻ cells in SKO liver (***Figure 7—figure supplement 3C***). Principal component analysis (PCA) of the IEG expression profiles showed overlap of all samples with each other, instead of partitioning into distinct groups, suggesting no biased expression of specific IEGs in the CD133⁺ cell population in SKO liver (***Figure 7—figure supplement 3D***). Nonetheless, the clustering of CD133⁺ cells in the center of the PCA plot showed that despite the stochastic expression pattern, CD133⁺ cells were featured by relatively similar and stable levels of IEG species. By measuring the overall expression amounts of IEGs in the selected list, we found that the total IEG levels were higher in WT than SKO cells at 4 hr after PHx (***Figure 7H***). We next analyzed IEG diversity within individual cells and variations in the expression among cells. For a fair comparison, the analysis was focused on Cyclin D1⁺ cells in the periportal area, to avoid influences of cell cycle and zonation in the liver (***Figure 7—figure supplement 3E***, and Methods). Indeed, the intracellular IEG diversity was high and intercellular variations were low among CD133⁺ hepatocytes, relative to other groups (***Figure 7I, J***). This data distinguished CD133⁺ cells of SKO liver from WT hepatocytes and also revealed that the increased entropy of IEGs was not due to the restoration of a WT-like expression pattern (***Figure 7G***). A plot with SKO hepatocytes showed clearly higher IEG diversity in CD133⁺ than CD133⁻ cells, with similar total IEG expression levels (***Figure 7K***). Consistent with the significance of the selected IEGs for cell proliferation, we observed a positive correlation between Cyclin D1, IEG diversity, and the total IEG levels (***Figure 7K***). When simulating mRNA exchanging, the plot pattern of CD133⁻ cells in the SKO liver moved toward a striking resemblance to the CD133⁺ population, increasing IEG diversity within each cell (***Figure 7L, M***). Mathematical modeling demonstrated that exchanging 1/12 of IEGs in a cell with four other cells twice was sufficient to reach the distinctive IEG profile of CD133⁺ cells (***Figure 7L, M***, ***Figure 7—figure supplement 4A–4C***, and Methods). Based on the cellular and RNA-seq data, together with math modeling, we propose that reciprocal mRNA sharing within clustered CD133⁺ cells may drive an enhanced proliferative phenotype, despite signaling deficiency in the whole population.

## CD133 is necessary to sustain cell proliferation under signal deficit

Given the compensatory CD133 upregulation, we reasoned that a defective phenotype associated with CD133 loss could be amplified in a compound mutant also deficient for a pro-proliferative molecule. To test this theory, we crossed the *Prom1* KO mouse with a conditional *Shp2^flox/flox* mouse line, and deleted Shp2 in hepatocytes by AAV-Cre virus infection. At 2 days after PHx, hepatocyte proliferation in E-Cad⁺ colonies was significantly suppressed in the Prom1/Shp2 double KO (DKO) hepatocytes, compared to SKO control (***Figure 8A, B***), although Prom1 KO alone did not inhibit Shp2⁺ hepatocyte proliferation (***Figure 8B***). We also isolated hepatocytes from SKO and DKO (Prom1 KO; *Shp2^flox/flox*; Alb-Cre) livers, and examined cell colonies formed in culture. Cell proliferation in E-Cad⁺ colonies was markedly lower in DKO than in SKO hepatocytes (***Figure 8C, D***). However, the basal low levels of proliferation were similar between SKO and DKO non-colonized cells. We also observed higher proliferation rates of DKO hepatocytes within E-Cad⁺ colonies than non-colonized cells in culture, suggesting that CD133 is unlikely the sole mediator of the intercellular communication event. Next, we labeled SKO hepatocytes by injecting AAV-GFP into SKO mice, and mixed the GFP-labelled SKO cells with DKO hepatocytes at a 1:10 ratio in culture (***Figure 8E***). If CD133 had a cell-autonomous

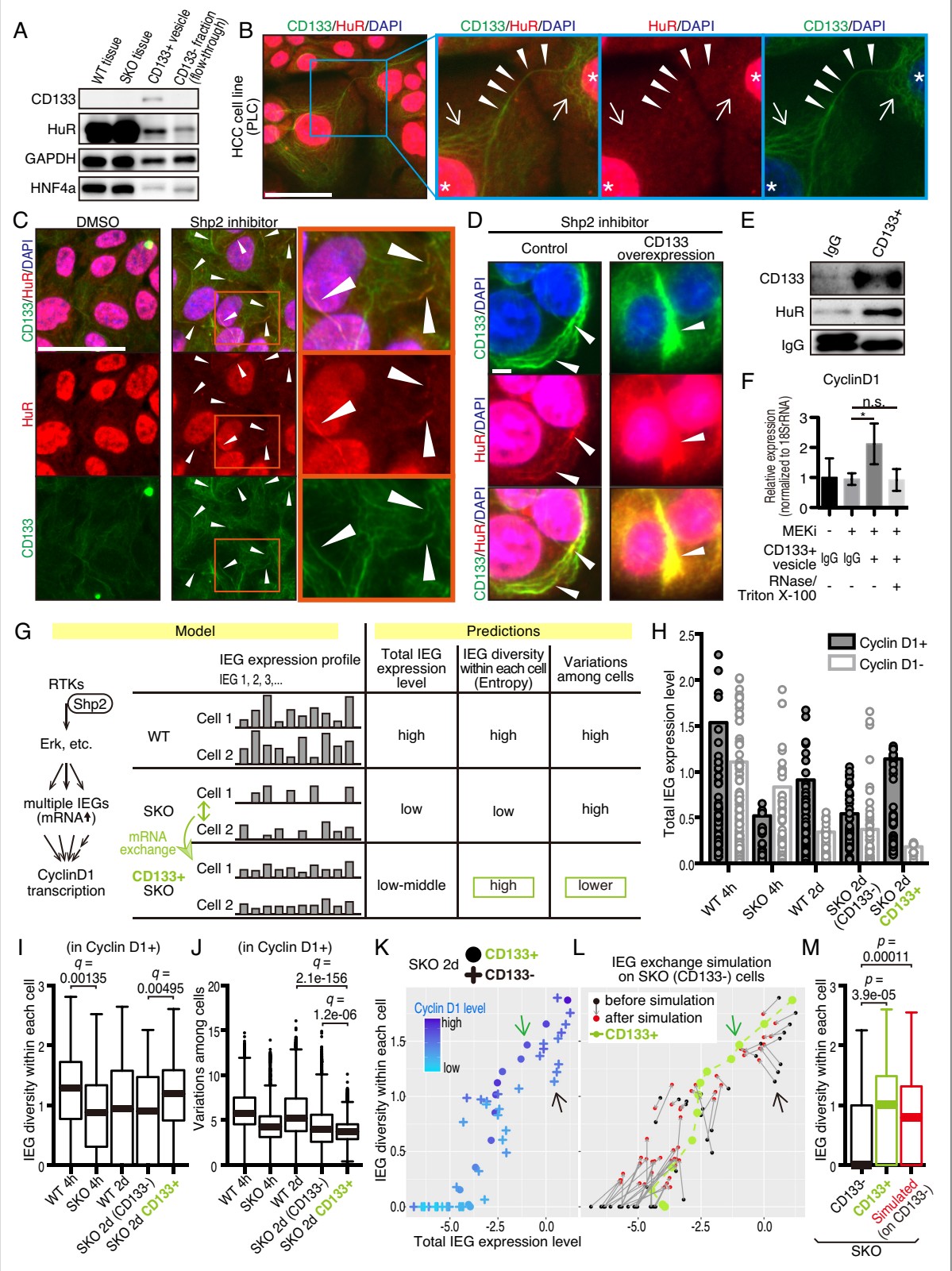

**Figure 7.** CD133-mediated mRNA sharing converts intercellular heterogeneity into intracellular diversity. (**A**) Immunoblotting of HuR in the CD133⁺ vesicles from Shp2 knockout (SKO) liver after partial hepatectomy (PHx). GAPDH and HNF4α were used as controls for cytoplasmic and nuclear fractions, respectively. (**B**) Immunofluorescence on PLC cells. Arrows, peri-nuclear areas enriched with HuR in the cytoplasm; Arrowheads show colocalization of HuR on the CD133⁺ filament bridging two cells. (**C**) Immunofluorescence on PLC cells treated with the Shp2 inhibitor (SHP099).

*Figure 7 continued on next page*

*Figure 7 continued*

Arrowheads show strong localization of HuR on the CD133$^+$ filaments. (**D**) Immunofluorescence on PLC cells transfected with CD133 expression vector treated with a Shp2 inhibitor (SHP099). Arrowheads show strong localization of HuR on the CD133$^+$ filaments. (**E**) Immunoblotting of CD133$^+$ vesicles isolated from MEK inhibitor-treated PLC cells. (**F**) qRT-PCR analysis of PLC cell lysates after treatment with CD133$^+$ vesicles isolated from MEK inhibitor-treated PLC cells. RNase and Triton X-100 were used to digest the RNA content of the vesicles. *$p<0.05$ (two-tailed unpaired t-test). n.s., not statistically significant. Means ± SD from three replicates are shown. (**G**) A model and predictions for single-cell RNA-seq data analysis. (**H**) Total immediate early-responsive gene (IEG) expression levels. Cyclin D1-positive and -negative cells were analyzed separately to evaluate the influence of the cell cycle on the IEG analysis. See the Methods section for what the bars and dots represent. (**I and J**) Box plots (Tukey's) of IEG diversity within each cell (calculated as entropy) and IEG variations among cells. Analyses were focused on cyclin D1$^+$ cells in all groups for fair comparison. (**K and L**) Plot of intracellular IEG diversity against total IEG expression levels in SKO hepatocytes 2 days after PHx. Blue color gradient indicates cyclin D1 expression levels. For the simulation in (**L**), the parameters used were: Group of 5 cells, X=1/12, model 3 (see also *Figure 7—figure supplement 4* and Methods). Green and Black arrows show typical profiles of CD133-positive and -negative cells, respectively. (**M**) Box plot (Tukey's) of intracellular IEG diversity after simulation. The analysis was not limited to cyclin D1-positive or -negative cells. Note the simulation of the IEG exchange attracted the cells from cyclin D1-low profile to cyclin D1-high profile (**K–M**). Statistics were performed by the Wilcoxon rank sum test adjusted by FDR in (**I**), (**J**), and (**M**). Scale bars, 50 μm (**B, C**), 5 μm (**D**).

The online version of this article includes the following source data, source code, and figure supplement(s) for figure 7:

**Source code 1.** Source code for the simulations in panels K-M.

**Source data 1.** Source data for western blots in panel A and E.

**Figure supplement 1.** Analyses of CD133$^+$ vesicles.

**Figure supplement 2.** Identification of hepatocytes in the single-cell RNA-sequencing (scRNA-seq) dataset.

**Figure supplement 3.** Single-cell RNA-seq analysis of WT and Shp2 knockout (SKO) livers after partial hepatectomy (PHx).

**Figure supplement 4.** Simulation analysis of the immediate early-responsive gene (IEG) exchange with single-cell RNA-sequencing (scRNA-seq) of the regenerating Shp2 knockout (SKO) liver.

function, the SKO cells in a DKO cell pool were expected to proliferate similarly as in a pure SKO cell pool. For example, if CD133 serves as a receptor, SKO cells in the mixture should still receive a ligand signal from DKO cells. By contrast, the SKO cells scattered within DKO colonies only proliferated as much as DKO cells (*Figure 8F, G*), arguing against a cell-autonomous role of CD133. CD133 removal in DKO cells did not abolish direct cell-cell contact, with tight cell adhesions maintained, as shown by E-cad staining (*Figure 8F*).

Given the abundant CD133 expression in the intestinal crypt, we isolated crypt cells from WT and Prom1 KO mice for organoid culture in vitro. Consistent with previous data (*Barker et al., 2010*), CD133 was expressed in intestinal crypt cells, likely including both the Lgr5$^+$ stem cells and the progenitor-like transit amplifying cells (*Figure 8—figure supplement 1*). Consistent with previous data, no significant difference in cell proliferation was observed between WT and Prom1 KO organoids (*Figure 8H, I*). However, treatment with Trametinib exhibited more prominent inhibition on cell proliferation in the crypt buds of Prom1 KO organoids than the WT control (*Figure 8H, I*), indicating higher sensitivity of CD133-deficient intestinal cells to impaired Ras-Erk signaling. Together, these data illustrate a functional role of CD133 in various cell types for strive to proliferate under signal deficit.

## Discussion

Originally aimed at deciphering hepatocyte proliferation in liver regeneration, this study has unveiled a cell-cell communication mechanism shared in various cell types under defective proliferative signaling. Interestingly, this was deciphered through the expression of CD133, a functionally mysterious molecular marker for stem cells and cancer stem cells. By taking multidisciplinary approaches, especially super-resolution microscopy, and scRNA-seq, we obtained data that suggest a role of CD133$^+$ vesicle for the intercellular exchange of mitogenic mRNAs. Although CD133 was thought to label stem cells in normal and tumor tissues, its expression is not restricted to the stem cell populations in either case. We found that CD133 expression was transiently induced in Shp2- and MET-deficient hepatocytes in regenerating livers damaged by PHx or CCl$_4$. Of note, the CD133 upregulation was not accompanied by an increase in other stem/progenitor cell markers or associated with cell lineage transition. Furthermore, the reversable CD133 induction was detected in several types of cancer cells following the

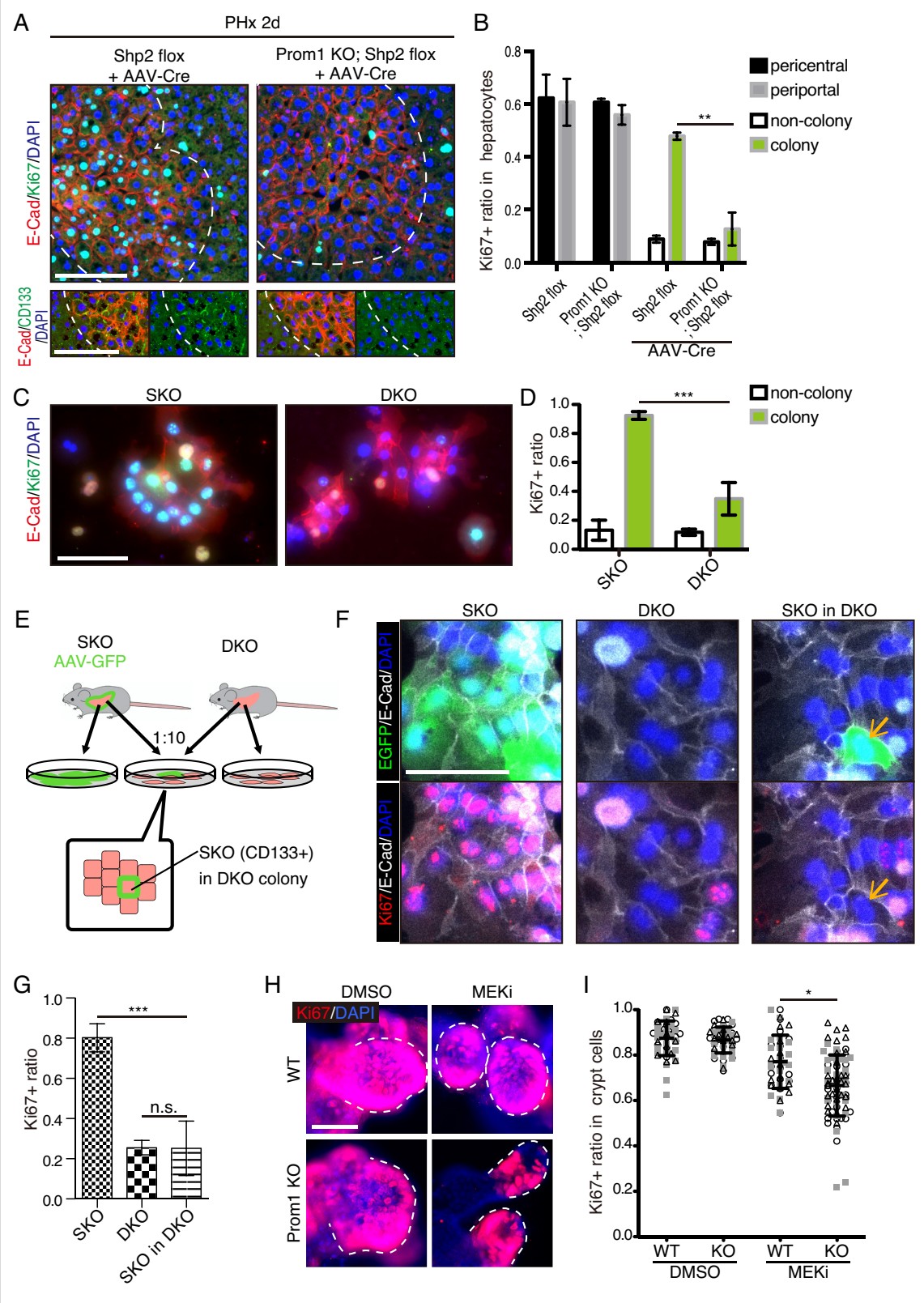

**Figure 8.** CD133 is required for cell proliferation under proliferative signal deficit. (**A and B**) Immunofluorescence (**A**) on liver sections of WT and Prom1 KO mice 2 days after partial hepatectomy (PHx) with or without Shp2 deletion using AAV-Cre and quantification of Ki67⁺ ratio in hepatocytes in the indicated genotypes (**B**). Separate analyses of pericentral and periportal hepatocytes showed insignificance of zonal difference. **p<0.01, (two-tailed unpaired t-test). Means ± SEM are shown. n=3, 3, 4, and 5 mice, respectively. (**C and D**) Immunofluorescence (**C**) on primary hepatocytes isolated from

*Figure 8 continued on next page*

*Figure 8 continued*

Shp2 knockout (SKO) and Shp2/Prom1 double KO (DKO) mouse livers and quantification of Ki67$^+$ ratio (**D**). Images of representative colonies are shown. ***p<0.001, (two-tailed unpaired t-test). Means ± SD from four wells are shown. (**E**) Experimental design with primary hepatocytes isolated from GFP-labeled SKO liver and unlabeled DKO liver. E-Cad$^+$ colonies were analyzed. (**F**) Immunofluorescence images of E-Cad$^+$ colonies in SKO, DKO, and mixed culture as shown in (**E**). The arrows show a GFP$^+$ SKO cell forming a part of the colony with the surrounding DKO cells. (**G**) Quantification of the Ki67 ratio in E-Cad$^+$ colony-forming cells is shown in (**F**). ***p<0.001 (two-tailed unpaired t-test). n.s., not statistically significant. Means ± SD from three wells are shown. (**H and I**) Immunofluorescence (**H**) on WT and Prom1 KO mouse intestinal organoids treated with MEK inhibitor (MEKi) and quantification of Ki67$^+$ ratio in the crypt cells (**I**). Dashed lines indicate the crypt buds. *p<0.05, (two-tailed unpaired t-test). Means ± SD from three wells are shown. Each dot represents each crypt buds, and symbols indicate each well. Scale bars, 100 µm (**A, C, and F**) and 50 µm (**H**).

The online version of this article includes the following figure supplement(s) for figure 8:

**Figure supplement 1.** CD133 is required for the proliferation of crypt cells in the intestinal organoids during signal deficit.

suppression of RTK-Ras-Erk signaling by chemical inhibition of Shp2 or Mek. These data on dynamic CD133 expression suggest that its expression pattern is not strictly associated with cell identity but rather is modulated by intra- and inter-cellular signals, which may clarify some conflicting data and notions documented in the literature. Given that proliferative signaling can be frequently disturbed in health and diseases, upregulated CD133 expression and function is likely a conserved mechanism for cell proliferation under stress in various cell types.

This study identified CD133-associated filaments that connect neighboring cells (*Figure 5*). Super-resolution microscopy and immuno-EM further demonstrated that CD133 was primarily located on intracellular vesicles. The CD133$^+$ vesicles may mediate intercellular sharing of materials between tightly contacting cells with minimal diffusion. This putative role of CD133 explains why CD133/Prom1 deletion did not affect general development in mice, as CD133$^+$ vesicles are mainly generated in stress responses. However, CD133 is unlikely the sole molecule responsible for the formation of the vesicles; other molecules may also participate in this important cellular event. It is also plausible that cells use a CD133-independent mechanism to cope with stress in the absence of CD133. Furthermore, this study did not exclude a possibility that CD133 has other functions in cellular activities. Although, we identified intracellular CD133$^+$ vesicles here, this molecule with five transmembrane domains may play a different role on the cell surface in a context that was not studied here. CD133$^+$ vesicles were reportedly detected in internal and external body fluids (*Marzesco, 2013*), although their cell origin and functional significance are unclear. It is conceivable that these reported CD133$^+$ vesicles were released from damaged cells rather than excreted by healthy cells.

Unlike exosomes or EVs (*Valadi et al., 2007*), CD133$^+$ vesicles are enriched with mRNAs rather than miRNAs, especially the IEG transcripts. Math modeling based on the scRNA-seq data suggests that exchanging mRNAs between CD133$^+$ cells could convert intercellular stochastic heterogeneity into higher intracellular diversity (entropy). Simulating the process with the math model reveals that reciprocal mRNA exchange among defective cells is sufficient to restore a proliferative capacity of CD133$^+$ SKO hepatocytes deficient for proliferative signaling. Further experimental evidence is required to validate or modify this intercellular mRNA exchange theory. The diversity of mitogenic mRNAs is likely more important than the upregulation of individual genes during cell proliferation. In this case, while the sharing of mRNAs to increase the entropy is used by non-stem cells for compensation of signal deficiency, stem cells may more often require this mechanism to maintain their functions, especially under stress. Indeed, recent scRNA-seq analyses suggest intracellular transcriptional diversity (entropy) as a common feature of stem cells (*Grün et al., 2016*; *Gulati et al., 2020*; *Teschendorff and Enver, 2017*). An intracellular transcriptional diversity can be calculated as Shannon's entropy or a count of gene transcript species within individual cells. Due to the higher entropy of mRNAs, stem cells can shift their identity via differentiation or maturation, during which the transcriptional diversity is lowered for specific cell lineages. This property may explain why stem cells are more resilient in general, as higher entropy makes a more robust buffering capacity against various disturbances. Therefore, the intracellular transcriptional diversity is not merely a signature of stem cells but is functionally required for such robustness. Despite common features of various stem cells, there are no universal molecules that can identify or define all types of stem cells. Massive genomic data analyses in the literature have revealed different sets of biomarkers expressed in various stem cells. However, the intracellular diversity is likely a common property of all stem cells as well as actively dividing non-stem cells, which is maintained at least in part by CD133$^+$ vesicles.

Despite the controversial concept of CSC and CD133 as its biomarker, many reports have documented a significant correlation between CD133 expression, drug resistance, and prognosis of cancer patients. The CD133$^+$ cancer cells are reportedly more resistant to radio/chemotherapies and thus likely responsible for tumor recurrence (*Bao et al., 2006*; *Bertolini et al., 2009*; *Ma et al., 2008*; *Sarvi et al., 2014*). We observed markedly increased expression of CD133 in several tumor cell lines following treatment with growth-inhibitory compounds, which are already used in clinical treatment or trials (*Figure 4*). These data suggest the plasticity of CD133 expression in the acute response of tumor cells to anti-tumor drugs, and a group of tumor cells may manage to proliferate through intercellular communications upon inhibition of proliferative signals. This swift response to drug treatment does not require new mutations and is unrelated to the CSC identity. If considering 'stemness' as a trait rather than an identity, it is likely that any cancer cells acquiring high transcriptional diversity may show some signs of stemness, such as drug resistance. The CD133 expression and the CD133$^+$ vesicles are induced transiently in various cancer cell types, independent of their differentiation status. Thus, this study may have disclosed a long-sought CD133 function in cancer, further characterization of which may clarify the controversial issues regarding CD133 and the so-called cancer stem cells.

Many questions remain to be answered for this new type of vesicles, including their biogenesis, structure, components (RNAs, proteins, and lipids), and functions. Although our data suggest direct traffic of CD133$^+$ vesicles between adjoining cells, it is unclear how these vesicles migrate from one cell to another, going through tunneling nanotube-like structures or through the protrusions at donor and recipient cell surface via budding and endocytosis. As they were not localized to multivesicular bodies, CD133$^+$ intercellsomes are unlikely originated from the same pathway as the exosomes. The exact function and mechanism of CD133 in intercellsomes are yet to be elucidated, although our data are in agreement with previous results showing the association of CD133 with cholesterol and lipid membrane morphology changes (*Röper et al., 2000*; *Thamm et al., 2019*). Answering these questions will likely elucidate a new intercellular communication mechanism associated with stress responses, under physiological and pathological conditions. Simultaneously suppressing intracellular proliferative signaling and disrupting the compensatory intercellsome function may be a new strategy to overcome drug resistance and tumor relapse.

## Materials and methods
### Animals
Hepatocyte-specific Shp2 knockout (*Alb-Cre; Shp2$^{flox/flox}$*) or SKO mice were generated as described previously (*Bard-Chapeau et al., 2011*). *c-Met$^{flox/flox}$* mice (FVB;129P2-*Met$^{tm1Sst}$*/J) were purchased from the Jackson Laboratory and bred with Alb-Cre mice to produce hepatocyte-specific c-Met KO mice. Prom1 KO mice (B6N;129S-*Prom1$^{tm1(cre/ERT2)Gilb}$*/J) were purchased from the Jackson Laboratory and bread with *Shp2$^{flox/flox}$* and SKO mice. *Shp2$^{flox/flox}$* littermates were used as WT control mice. All experimental procedures were approved by the UCSD Institutional Animal Care and Use Committee (IACUC). PHx was performed on mice at age of 8–10 weeks with the use of Buprenorphine hydrochloride (Par Pharmaceutical) as a pain relief for the surgery. CCl$_4$ (20% in 100 µl corn oil/20 g of body weight; Sigma) was intraperitoneally injected into 8–10 week-old mice. AAV-Cre (AAV8.TBG.PI.Cre.rBG) was purchased from the Penn Vector Core and injected into the tail vein (2x10$^{11}$/28 gbw). For chemical induction of hepatocellular carcinoma (HCC), DEN (25 mg/kg; N0258-1G; Sigma) was injected intraperitoneally into mice at postnatal day 15. BrdU (1 mg/mice; Sigma) was intraperitoneally injected twice a day for 3 days.

### Cell culture and experiments
PLC/PRF/5 cells were cultured in DMEM medium supplemented with 10% fetal bovine serum. The MCF10A cells were cultured in DMEM-F12 medium (Gibco) with 5% horse serum (Gibco) and 1% penicillin-streptomycin supplemented with 20 ng/mL human EGF (Gemini), 0.5 µg/mL hydrocortisone, 100 ng/mL cholera toxin (Sigma-Aldrich), and 10 µg/mL human insulin (Gemini). The cultures were maintained at 37 °C in a humidified environment containing 5% CO2. For inhibitors treatment, PLC/PRF/5 cells were treated with HGF and SHP099 (10 µM; CHEMIETEK; CT-SHP099) or trametinib (10 nM; APExBIO; A3018) for 2–6 days after cell plating and starvation overnight at the second day. MCF10A cells were treated with SHP099 (20 µM) or trametinib (10 nM) for 2–6 days after cell

plating. Medium was changed every two days. Cells were transfected with CD133-Myc-Tag expression vector (HG15024-CM; SinoBiological) using Lipofectamine 3000 transfection reagent (Invitrogen). For lentivector transfection, the fusion protein and fluorescent proteins were subcloned into pLJM1 vector. HEK293T cells were transfected with the lentivectors together with psPAX2 and pCMV-VSV-G, using Lipofectamine 3000 (Thermo Fisher Scientific). Supernatant was collected, centrifuged at 3000 g for 5 min, filtered through 0.45 µm filter, and used to transfect the PLC cells. The lentiCRISPR v2 was used for the CRISPR experiment.

## Immunostaining

Freshly frozen sections or fixed frozen sections were used for immunofluorescent staining. Freshly frozen sections and cultured cells were fixed by cold 4% PFA in PBS, cold acetone, or cold methanol for 30 min to overnight, depending on the antibodies used. Antigen retrieval was performed for some stainings for 10 min to 1 hr, depending on the antibody used. Antibodies for Ki67 (14-5698-80; eBioscience), HNF4α (sc-8987; Santa Cruz Biotechnology), mouse CD133 (14-1331-82; eBioscience), human CD133 (86781; CST), E-cadherin (sc-7870; Santa Cruz Biotechnology), GFP (04404–84; Nacalai Tesque), Shp2 (sc-280; Santa Cruz Biotechnology), Porcupine (ab105543; abcam), β-catenin (sc-7199; Santa Cruz Biotechnology), CHMP2B (ab157208; abcam), HuR (ab200342; abcam, and 66549–1-Ig; proteintech), VE-Cad (AF1002; R&D SYSTEMS), BrdU (555627; BD Biosciences), PECAM (13-0311-81; eBioscience), HGF (ab83760; abcam), GFAP (Z0334; Dako), EpCAM (552370; BD Biosciences), Sox9 (AB5535: Millipore), CD44v6 (GTX75661; GeneTex), and AFP (AF5369; R&D SYSTEMS) were used as primary antibodies. Fluorescent images were taken with a monochrome camera, and pseudo colors were applied for each channel. STORM images were taken with Nikon A1R TIRF STORM Microscope at the Moores Cancer Center. Cryo-ultramicrotome sectioning, fluorescence staining, immunogold staining, and electron-microscopy (EM) were performed by the Electron Microscopy Core Facility at UCSD.

## In situ hybridization

Freshly frozen sections of the regenerating SKO liver and PLC cell line treated with Meki were subjected to HCR RNA-FISH, using probes and reagents purchased from Molecular Instruments after the CD133 staining. Freshly frozen sections were fixed with PFA for 30 min, washed with PBS, and further fixed/permeabilized with EtOH overnight. PLC was fixed with MeOH overnight. The fixed sections and cells were then immunostained for CD133 using the antibodies described above. After the immunostaining, HCR RNA-FISH was performed according to the protocol provided by Molecular Instruments. ProtectRNA RNase Inhibitor (R7397; SIGMA) was added to all the reagents used. The images were acquired using Leica SP8 confocal microscopy.

## Primary hepatocyte isolation and culture

Each liver was perfused with 10 ml of HBSS (without $Ca^{2+}$ or $Mg^+$) containing 0.5 mM EDTA, 4 ml of HBSS (without $Ca^{2+}$ or $Mg^+$) and 10 ml of collagenase solution [2 mg/ml collagenase H (11074059001; Roche), 0.1 mg/ml DNaseI (10104159001; Roche), 5 mM $CaCl_2$ and 0.9 mM MgCl in HBSS] sequentially. All the solutions were pre-warmed to 37 °C before use. Livers were dissected and further incubated in 7 ml of collagenase solution at 37 °C for 5 min and dissociated. Dissociated cells were passed through a cell strainer and centrifuged at 50 g for 5 min to precipitate hepatocytes. For qRT-PCR, the pellet was immediately dissolved in Trizol for the hepatocyte fraction. The supernatant containing NPCs was washed with PBS at 50 g for 5 min three times to eliminate hepatocytes, and precipitated at 400 g for 5 min, and dissolved in Trizol, and used as NPC fraction. For hepatocyte culture, the hepatocyte pellet after the first centrifugation was resuspended and washed with PBS at 50 g for 5 min three times to eliminate NPCs, resuspended in the culture media [Williams' Media E supplemented with Primary Hepatocyte Maintenance Supplements (CM4000; Gibco) as instructed] and seeded on collagen-coated plates.

## Intestinal organoid culture

After dissecting the intestinal tube from the mice, the tubes were washed with PBS, by flushing with a syringe. The villi were gently scraped off and the tissues were cut into smaller pieces. The tissue fragments were rinsed with PBS three times, and then incubated in PBS with 1 mM EDTA for 30 min.

After the incubation, the fragments were put in fresh PBS and rigorously shaken, and the cells/debris that came off were discarded. The tissue fragments were then incubated in PBS with 5 mM EDTA for 1 hr. After the incubation, the fragments in the solution were rigorously shaken and pipetted, and the cell clumps that came off were used as the isolated crypts. The crypts were passed through 100 µm cell strainer, and spun down and washed with PBS at 300 x g for 10 min three times. The isolated crypt fragments were embedded in Matrigel and cultured in DMEM/F12 containing B27 (17504044; Gibco), N2 supplement (17502048; Gibco), 1.25 mM n-Acetylcysteine, 50 ng/ml EGF, 100 ng/ml Noggin, and 500 ng/ml R-spondin 1. Organoids were passaged in the same media, and treated with 50 nM trametinib.

## Immunoblotting

Proteins were extracted in RIPA buffer, and immunoblotting was performed using standard protocols. In addition to antibodies used for immunostaining, antibodies for EpCAM (bs-1513R; Bioss), HNF4α (GTX89532; GeneTex), GAPDH (5174; CST), β-actin (A5316; Sigma), Stat3 (9132; CST), Erk1/2 (4695; CST), and Cyclin D1 (sc-20044; Santa Cruz Biotechnology) were used as primary antibodies.

## Hydrodynamic tail vein injection

Plasmids pT3-elongation factor 1α (EF1α)-Shp2 (5 µg/ml), pT3-EF1α-GFP (5 µg/ml), p-cytomegalovirus (pCMV)-sleeping beauty (SB) transposase (1.5 µg/ml), or were diluted in PBS. Total volume of 2 ml/20 g.b.w. was rapidly injected into the tail vein. The GFP-expressing vector has transposon sequences, and when injected together with the SB transposase-expressing vector, the GFP-expressing sequence was integrated into the genome. Plasmids were prepared using the GenElute HP Endotoxin-Free Plasmid Maxiprep Kit (NA0410; Sigma). Silica fines in the final product were carefully eliminated by additional centrifugation at 12,000 g for 30 min.

## Isolation of CD133+ vesicles

Liver perfusion was similar to hepatocyte isolation as described above, except that collagenase Type IV (17104019; Gibco) was used instead of collagenase H to minimize unspecific digestion of proteins other than collagen. The livers were dissected, immersed in the collagenase solution, and dissociated and mildly lysed. Cells and debris were removed by centrifugation at 4000 g for 10 min three times, followed by filtration with 0.45 µm PES filter. We obtained around 2 ml of solution per mouse at this point. EDTA was added to suppress the collagenase activity and sodium azide was added to prevent possible degradation by bacterial growth. For PLC cells, cells were lysed in PBS by Dounce homogenizer, and cells and debris were removed similarly. FITC conjugated anti-CD133 antibody (1:500 dilution; 11-1331-82; Invitrogen) was added, and the solution was incubated at 4 °C overnight. Objects labeled with anti-CD133 antibody were then targeted with magnetic particles, using the FITC selection cocktail (18558; STEMCELL TECHNOLOGY), and isolated using MACS MS column (130-042-201; Miltenyi Biotec). CD133+ fraction was directly eluted with either lysis buffer for immunoblotting or lysis buffer for RNA isolation using RNeasy Plus Micro Kit (74034; QIAGEN). Isolation protocol for total RNAs including small RNAs was used according to the manufacturer's instruction with the RNeasy Plus Micro Kit. For the agarose gel electrophoresis, total RNAs were normalized to the same concentrations, mixed with RNA loading dye (final concentration of 47.5% formamide, 0.01% SDS, 0.01% bromophenol blue, 0.005% Xylene Cyanol FF, 0.01% SYBR safe, 0.5 mM EDTA), denatured at 65 °C for 15 min, cooled on ice, and loaded to the gel. For the treatment of cells with the isolated vesicles, Anti-PE MultiSort Kit (130-090-757; Miltenyi) was used instead of the FITC selection cocktail, to enzymatically detach the vesicles from the magnetic beads. As the vesicles were not likely secreted and endocytosed naturally, we forced the cells to take the vesicles by using Lipofectamine 3000.

## Isolation of extracellular vesicles

Extracellular vesicles were isolated from the culture supernatant using Total Exosome Isolation Reagent (Invitrogen). Debris or cell contaminants were removed by centrifugation, larger vesicles such as apoptotic bodies were subsequently removed by filtration, and extracellular vesicles were precipitated according to the manufacturer's instructions. Total RNAs were extracted in the same way as the extraction from CD133+ vesicles described above.

## qRT-PCR

RNAs extracted by either Trizol or RNeasy Plus Micro Kit were reverse transcribed to cDNAs using High-Capacity cDNA Reverse Transcription Kit (4368814; Applied Biosystems), and quantitative PCR was performed using DyNAmo Flash SYBR Green qPCR Kit (F415S; Thermo Scientific). Primers for human CD133 (Forward GCGTGATTTCCCAGAAGATA; Reverse CCCCAGGACACAGCATAGAA), human EpCAM (Forward CCATGTGCTGGTGTGTGAA; Reverse TGTGTTTTAGTTCAATGATGATCCA), human Sox9 (Forward GTACCCGCACTTGCACAAC; Reverse TCTCGCTCTCGTTCAGAAGTC), human NANOG (Forward TCTCCAACATCCTGAACCTCA; Reverse TTGCTATTCTTCGGCCAGTT ), human CD44 (Forward GCAGTCAACAGTCGAAGAAGG; Reverse TGTCCTCCACAGCTCCATT), human Myc (Forward CCTTCTCTCCGTCCTCGGAT; Reverse CTTGTTCCTCCTCAGAGTCGC), human Jun (Forward GAGCTGGAGCGCCTGATAAT; Reverse CCCTCCTGCTCATCTGTCAC), and human CyclinD1 (Forward GCTGTGCATCTACACCGACA; Reverse TTGAGCTTGTTCACCAGGAG) were used for human cell lines.

## Single-cell RNA library construction and sequencing

Single hepatocytes were isolated from WT at 4 hr (4 hr) and 2 days (2d), SKO at 4 hr and 2d, and DKO at 2d after PHx, by a two-step in-situ perfusion with collagenase H as described above. To remove dead cells and debris, the pelleted hepatocytes were resuspended in 45% Percoll (sigma-aldrich) and centrifuged at 50 g for 10 min. Cells were further washed with PBS and counted with a hemo-cytometer. The isolated hepatocytes were loaded onto a 10 x Chromium Controller and then parti-tioned into nanoliter-scale Gel Beads-In-Emulsion (GEMs). The volume of single-cell suspension was calculated in order to generate 5000 GEMs per sample. Libraries were constructed with Chromium Single Cell 3' Reagent Kits (v3 chemistry, 10 x Genomics). Sequencing was performed using HiSeq 4000 (Illumina) at IGM Genomics Center, University of California, San Diego, with the following read length: Read 1, 28 bp, including 16 bp cell barcode and 12 bp unique molecular identifier (UMI); Read 2, 98 bp transcript insert; i7 sample index, 8 bp. For the single-cell RNA-seq data, GEO accession number: GSE169320.

## Single-cell RNA-seq data preprocessing

The paired reads from WT and SKO samples were aligned to mouse reference genome GRCm38 using CellRanger package (v3.0.2). And the reads from DKO were aligned to modified GRCm38, with Prom1-CreERT2 sequence manually added. An expression matrix, with each row representing a gene and each column representing a cell, was generated for each sample, and then filtered for downstream analysis. In brief, for each sample, genes expressed in less than three cells were removed; cells failed to meet the following criteria were removed: (1) the number of genes detected in each cell should be over 200 and less than 5000; (2) the UMIs of mitochondrial genes should be less than 20% of total UMI; (3) the UMIs of a cell should be less than 20,000 to avoid non-singlet (i.e. transcriptome representing more than one cell).

## Detection of cell subpopulations

To identify subpopulations within hepatocytes and select clusters for further analyses, we examined each sample using R package Seur(v2.4) (*Butler et al., 2018*). First, the raw expression matrix was normalized by the total expression, multiplied by a scale factor of 10,000, and log-transformed. Next, we regressed on the number of UMIs per cell as well as mitochondrial gene percentage. After normalization and scaling, z-scored residuals were stored for dimensionality reduction and clustering (*Figure 7—figure supplement 2*). The hepatocytes showed high expression of *Hnf4a* and *Asgr1*. And other clusters with expression of *Ptprc*, *Lyve1*, *Pecam1*, *Lrat*, *Hba-a1*, and *Krt19* (contaminant immune cells, endothelial cells, hepatic stellate cells, and erythrocytes) were removed. Next, for hepatocytes in each sample, PCA was performed first, and 75 PCs were used for tSNE analysis with other param-eters, including perplexity = 30 and maximum iteration = 2000. To find hepatocyte subpopulations, the same PCs were imported into FindClusters from Seurat. All samples were also colored by *Alb* (marker for hepatocytes), *Cyp2f2* (marker for periportal hepatocytes), and *Cyp2e1* (marker for peri-central hepatocytes) expression. CD133[+] cells of SKO liver (2d after PHx) were not limited to, but abundantly identified in one of the two clusters of periportal hepatocytes (80%), which was consistent with our observations on tissues of the PHx model. After comparing differentially expressed genes in

two periportal clusters, we found the cluster enriched with CD133$^+$ cells in the SKO 2d sample, had higher *Alb* and *Hamp* expression, presumably reflecting a population that is located in a particular zone within the periportal area. The same population (*Alb*$^{high}$, *Hamp*$^{high}$, *Cyp2f2*$^+$) was also observed in other samples. For a fair comparison, further analysis was limited to this periportal population to avoid variances due to spatial locations within the liver tissue.

## Calculation of total IEGs' expression levels

To obtain the IEG list for analysis, we checked data in a public database (http://software.broadinstitute.org/gsea/msigdb/index.jsp) and publications (*Locker et al., 2003*; *Su et al., 2002*), for genes expressed during liver regeneration or after liver injury. Among the genes upregulated, we eliminated the genes that were irrelevant to cell growth or those acting as negative feedback regulators. As a result, we obtained 12 IEGs for further analyses, including *Fos, Jun, Egr1, Ier2, Atf3, Junb, Myc, Crem, Ets2, Ier3, Lepr, Egfr*. To integrate the total expression level of the 12 genes, we defined a score based on the zero-inflated Negative Binomial (ZINB) model. ZINB regression is for modeling count variables with excessive zeros. It was first introduced to model single-cell RNA-seq data by *Miao et al., 2018*, and it had shown high accuracy in differential expression analysis compared to other models (*Wang et al., 2019*). The score for total IEGs expression was computed as follows. (1) Within a group of cells, we first fitted counts of each gene by a ZINB model:

$$
\begin{aligned}
f_{ZINB}(n; \theta, r, p) \equiv P(N - n) \quad &= \theta \cdot I(n = 0) + (1 - \theta) \cdot f_{NB}(n : r, p) \\
&= \theta \cdot I(n = 0) + (1 - \theta) \cdot \binom{n = r - 1}{n} p^n (1 - p)^r
\end{aligned}
$$

where θ is a proportion of 'real' zeros in this group of cells ('real' zeros represent that the gene was not expressed in some cells, instead of the dropout caused by mRNA capture procedure); $r$ and $p$ were parameters from the negative binomial distribution, representing size and probability, respectively; $n$ is the median normalized count (*Miao et al., 2018*) of genes. ZINB model connects 'real' zeros and a negative binomial model with θ and indicator function $I$. The observed excessive zeros from single-cell RNA-seq dataset might come from either 'real' zeros or zeros in the negative binomial model. (2) Therefore, the expected count or mean for each gene was calculated as follows:

$$
\begin{aligned}
\mu_{ZINB} \quad &= P(\text{gene not expressed}) \cdot 0 + P(\text{gene expressed}) \cdot \mu_{NB} \\
&= (1 - \theta) \cdot \frac{pr}{1 - p}
\end{aligned}
$$

For each group of cell $c$ and each gene $g$, we calculated the expected count as $\mu_{c,g}$. With the same method, the expected count of gene $g$ among all cells used for this analysis was calculated as $\mu_{a,g}$. (3) For cell cluster $c$, the score of total IEGs expression level was defined as:

$$
S_C = 2^{\frac{1}{N_g} \sum_g \log_2 \left( \frac{\mu_{c,g}}{\mu_{a,g}} + 0.001 \right)}, \quad \text{where } N_g = 12
$$

For a single-cell in a group, $a$, for example, we calculated its binary distance to all other cells and selected the 20 closest cells as a subgroup, called $c_a$. The score for $c_a$ was calculated following steps (1)–(3) as described above and plotted as a dot in *Figure 7H*.

## IEG diversity within each cell and variation among cells

We first used Shannon's entropy of all cells in a group to indicate the IEG diversity. The entropy $E_j$ of cell $j$ was computed as:

$$E_j = -\sum_{i=1}^{N} p_{i,j}\log_2 P_{i,j}$$

$$p_{i,j} \equiv r_{i,j}/\sum_i r_{i,j}$$

$$r_{i,j} \equiv count_{i,j}/\mu_i$$

where $count_{i,j}$ represents the median normalized count of gene $i$ in cell $j$, and $\mu_i$ equals the expected count among all cells used for analysis. The higher entropy defined above practically reflects (1) more species of IEGs are expressed by a cell, as well as (2) less overall deviation from mean expression values of IEGs within a cell. To describe the variation among cell subpopulations, we used cell pair-wise Euclidean distance with normalized and scaled data by Seurat. However, in order to do a fair comparison between samples when using scaled data, we first integrated four datasets by Canonical Correlation Analysis (CCA) with Seurat, before calculating the distance. Finally, the significance of diversity and variation between cell groups was decided by the Wilcoxon rank sum test and adjusted by FDR.

## RNA exchange simulation

We plotted IEG diversity (entropy) against total IEG expression levels in the SKO-2d sample. The cells were first divided into CD133⁻ and CD133⁺ groups. Within each group, all cells were ordered by their entropies from low to high. Starting from the first cell, we applied a moving average to entropies, with a sliding window of 20 cells and a step size of 5. Within each window, we also computed the total IEG expression value as described above. We then plotted IEG entropy against the total expression level for CD133⁻ and CD133⁺ groups. Every sliding window with 20 single-cells was treated as a meta-cell. Next, we examined how the cells would behave in the plot, when simulating the RNA exchange among CD133⁻ cells. The basic idea was to randomly divide all CD133⁻ cells into small groups with size N and make cells within the same group exchange IEGs with each other. We tested four possible RNA exchange models (*Figure 7—figure supplement 3A*):

- Model 1: Within a small group of size N, for each IEG, the cell with the highest expression gives X of its RNA to each of the other (N-1) cells; the corresponding RNA level is reduced by (N-1)*X in the donor cell.

$$\begin{cases} N = 3, 5, 7 \\ X = \dfrac{1}{16}, \dfrac{1}{12}, \dfrac{1}{8}, \dfrac{1}{6}, \dfrac{3}{16}, \dfrac{1}{4} \end{cases}$$

- Model 2: Within a small group of size N, for each IEG, RNA is transferred as described in Model 1, but the RNA level is maintained in the donor cell because of its continuous transcription.

$$\begin{cases} N = 3, 5, 7 \\ X = \dfrac{1}{16}, \dfrac{1}{12}, \dfrac{1}{8}, \dfrac{1}{6}, \dfrac{3}{16}, \dfrac{1}{4} \end{cases}$$

- Model 3: Within a small group of size N, for each IEG, there are two rounds of RNA exchange. Considering that the CD133 expression was already detected (in RNA level) at 4 hr after PHx (*Figure 7—figure supplement 2B*), this is likely the reasonable model, as they should have multiple chances of mRNA exchange while they change their profile along the time until 2 days after PHx. The first round is the same as Model 1: recipients get $X_1$ of the RNA in a donor cell and the donor cell loses (N-1)*$X_1$. After the first round of exchange is finished, the cell with the highest expression now is designated as a new donor. During the second round, the new donor loses (N-1)*$X_2$ of its RNA and each of the other (N-1) cells get $X_2$.

$$\begin{cases} N = 3, 5, 7 \\ X_1 = \dfrac{1}{16}, \dfrac{1}{12}, \dfrac{1}{8}, \dfrac{1}{6}, \dfrac{3}{16}, \dfrac{1}{4} \\ X_2 = \dfrac{1}{16}, \dfrac{1}{12}, \dfrac{1}{8}, \dfrac{1}{6}, \dfrac{3}{16}, \dfrac{1}{4} \end{cases}$$

- Model 4 (exchange all): Within a small group of size N (N=3, 5, 7), for each IEG and each cell, the RNA levels become the same, which is equal to the mean values before exchange.

After the RNA exchange simulation was applied, the IEG diversity (entropy) and total IEG expression were re-calculated in order to track their changes. For *Figure 7M*, the IEG diversity was calculated as described above, and the analysis included both Cyclin D1-negative and -positive cells.

## Bulk RNAseq sample preparation and data processing

Total RNAs extracted from vesicles as described above were subjected to ribosome-depleted total RNA library prep (Illumina) at UCSD Genomics Core. Sequencing was performed at UCSD Genomics Core on the NovaSeq 6000 platform. Raw reads data quality control was performed using fastqc, sequenced reads were mapped to the hg19 reference genome using STAR, and the number of mapped reads to each gene was counted by htseq-count. The R package DESeq2 was then used to normalize raw reads and perform differential expression analysis between CD133$^+$ vesicles vs whole cell samples, as well as EVs vs whole cell samples. The significantly expressed genes were determined with adjusted p-value <0.05.

## Heatmap

For single-cell RNA-seq, we randomly selected 54 cells from the same periportal cluster as described above, to make the heatmap. The raw counts were log-transformed with pseudo-count=1. For bulk RNA-seq, we made two heatmaps to compare gene expression between CD133$^+$ vesicles and EVs. In *Figure 6D*, the variance stabilizing transformed values of interested genes from CD133$^+$ vesicles and EVs were extracted and scaled. In *Figure 6—figure supplement 1E*, the transformed values from CD133$^+$ vesicles and EVs were further standardized to whole cells.

## Statistical analysis

All statistical analyses of data are described in the corresponding figure legends. All statistical analyses were performed with GraphPad Prism 7, except R was used for the single-cell RNA-seq analyses. Sample sizes were determined to meet the standard in the field, and no data were excluded. Covariates such as age and sex were randomly selected within the predetermined range. Most of the phenotypes in the study were so obvious that we could tell the experimental groups by just looking at the samples, even without being told. Nevertheless, blinding was performed whenever possible.

# Acknowledgements

We thank Drs. SE Wang, WS Chen, G Castillon, and Feng lab members for reagents and helpful discussion. scRNA-seq was conducted at the IGM Genomics Center, UCSD. Some images were taken at the UCSD School of Medicine Microscopy Core (supported by the grant NINDS P30NS047101), super-resolution microscopy was performed at Moores Cancer Center Microscopy Core (supported by the grant P30CA23100-28), and cryo-ultramicrotome and electron microscopy were performed at Cellular and Molecular Medicine Electron Microscopy Core (UCSD-CMM-EM Core, RRID: SCR_022039). The UCSD-CMM-EM Core is partly supported by the National Institutes of Health Award number S10OD023527. AS is funded by the Deutsche Forschungsgemeinschaft (DFG, German Research Foundation) –Projektnummer 502688960. This work was supported by NIH grants (R01DK128320, R01CA236074, and R01CA239629) to GSF.

---

# Additional information

### Funding

| Funder | Grant reference number | Author |
|---|---|---|
| National Institutes of Health | R01DK128320 | Gen-Sheng Feng |
| National Institutes of Health | R01CA236074 | Gen-Sheng Feng |
| National Institutes of Health | R01CA239629 | Gen-Sheng Feng |

| Funder | Grant reference number | Author |
|---|---|---|
| Deutsche Forschungsgemeinschaft | 502688960 | Alexander Scheiter |
| Moores Cancer Center, UC San Diego Health | Molecular Biology Cancer Fellowship | Kota Kaneko |

The funders had no role in study design, data collection and interpretation, or the decision to submit the work for publication.

## Author contributions
Kota Kaneko, Conceptualization, Data curation, Formal analysis, Supervision, Validation, Investigation, Visualization, Methodology, Writing – original draft, Writing – review and editing; Yan Liang, Data curation, Software, Formal analysis, Validation, Investigation, Visualization, Methodology, Writing – review and editing; Qing Liu, Shuo Zhang, Dan Song, Validation, Investigation; Alexander Scheiter, Funding acquisition, Validation, Investigation; Gen-Sheng Feng, Conceptualization, Resources, Data curation, Supervision, Funding acquisition, Writing – original draft, Project administration, Writing – review and editing

## Author ORCIDs
Kota Kaneko http://orcid.org/0009-0001-8595-8220
Gen-Sheng Feng http://orcid.org/0000-0003-1255-4708

## Ethics
This study was performed in strict accordance with the recommendations in the Guide for the Care and Use of Laboratory Animals of the National Institutes of Health. All of the animals were handled according to approved institutional animal care and use committee (IACUC) protocols (#S09108) of the University of California, San Diego.

Reviewer #1 (Public Review): https://doi.org/10.7554/eLife.86824.3.sa1
Reviewer #2 (Public Review): https://doi.org/10.7554/eLife.86824.3.sa2
Author Response https://doi.org/10.7554/eLife.86824.3.sa3

# Additional files

## Supplementary files
• MDAR checklist

## Data availability
Sequencing data have been deposited in GEO under an accession code GSE169320. Source data of the western blots are provided with this paper and source code for generating the simulations in Figure 7K-M is provided as Figure 7-source data 1.

The following dataset was generated:

| Author(s) | Year | Dataset title | Dataset URL | Database and Identifier |
|---|---|---|---|---|
| Kaneko K, Liang Y, Liu Q, Chen WS, Feng G | 2023 | CD133+ Intercellsome Mediates Direct Cell-Cell Communication to Offset Intracellular Signaling Deficit | https://www.ncbi.nlm.nih.gov/geo/query/acc.cgi?acc=GSE169320 | NCBI Gene Expression Omnibus, GSE169320 |

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
