## [Editor Report · eLife assessment]

This **important** study was designed to examine the bypass of Ras/Erk signaling defects that enable limited regeneration in a mouse model of hepatic regeneration. This hepatocyte proliferation is associated with the expression by groups of cells of mRNA-loaded CD133+ intracellular vesicles that mediate an intercellular signaling pathway that supports proliferation. These are new observations, supported by **convincing** data, that have broad significance to the fields of regeneration and cancer.

---

## [Referee Report · Reviewer #1 (Public Review)]

This study was designed to examine the bypass of Ras/Erk signaling defects that enable limited regeneration in a mouse model of hepatic regeneration. The authors show that this hepatocyte proliferation is marked by expression of CD133 by groups of cells. The CD133 appears to be located on intracellular vesicles associated with microtubules. These vesicles are loaded with mRNA. The authors conclude that the CD133 vesicles mediate an intercellular signaling pathway that supports cell proliferation. These are new observations that have broad significance to the fields of regeneration and cancer.

The primary observation is that the limited regeneration observed in livers with Ras/Erk signaling defects is associated with CD133 expression by groups of cells. The functional significance of CD133 was tested using Prom1 KO mice - the data presented are convincing.

The major weakness of the study is that some molecular mechanistic details are unclear - this is, in part, due to the extensive new biology that is described. Nevertheless, the data used to support some key points in this study are unclear:

a) What is the evidence that the observed CD133 groups of cells are not due to clonal growth. Is this conclusion based on the time course (the groups appear more rapidly than proliferation) or is this based on the GFP clonal analysis?

b) What is the evidence that the CD133 vesicles mediate intercellular communication. This is an exciting hypothesis, but what is the evidence that this happens? Is this inferred from IEG mRNA diversity? or some other data. Is there direct evidence of transfer - for example, the does the GFP clonal analysis show transfer of GFP that is not mediated by clonal proliferation? Moreover, since the hepatocytes are isogenic, what distinguishes the donor and recipient cells?

Increased clarity concerning what is hypothesis and what is directly supported by data - would improve the presentation of this study.

---

## [Referee Report · Reviewer #2 (Public Review)]

The manuscript by Kaneko set out to understand the mechanisms underlying cell proliferation in hepatocytes lacking Shp2 signals. To do this, the authors focused on CD133 as the proliferating clusters of cells in the Shp2 knockout (SKO) livers are CD133 expressing. After excluding the contribution of progenitors that are CD133 to this cell population, the authors focused on the intrinsic regulation of CD133 by Met/Shp2 regulated Ras/Erk parthway and showed upregulation of CD133 to be a compensatory signal to overcome loss of Ras/Erk signal and suggested Wnt10a in the regulation of CD133 signal. The study then focused on the observed filament localization of CD133 in the CD133+ cluster of cells. The study went on to identify the CD133+ vesicles that contain primarily mRNA vs. microRNA like other EVs. Specifically, the authors identified several mRNA species that encode IEGs, indicating a potential role for these CD133+ vesicles in cell proliferation signal transmission to neighboring cells via delivery of the IEG mRNAs as cargos. Finally, they showed that the induction of CD133 (and by derivative, the CD133+ vesicles) are necessary for maintaining cell proliferation in the cell cluster with high proliferation capacities in the SKO livers; and in intestinal crypt organoids treated with Met inhibitors to block Ras/ERk signal. In the revised manuscript, the authors more definitively identified the CD133+ vesicles. The authors also provided additional experimental evidence demonstrating the role of these CD133+ vesicles in cell-cell communication. The functional significance of CD133 on this cell-cell communication was further demonstrated with genetic knockout studies.

---

## [Author Response]

The following is the authors’ response to the original reviews.

**eLife assessment**
This important study was designed to examine the bypass of Ras/Erk signaling defects that enable limited regeneration in a mouse model of hepatic regeneration. This hepatocyte proliferation is associated with the expression by groups of cells of mRNA-loaded CD133+ intracellular vesicles that mediate an intercellular signaling pathway that supports proliferation. These are new observations, supported by convincing data, that have broad significance to the fields of regeneration and cancer.

First of all, we greatly appreciate the very positive take of this work by eLife editors and also thank the two reviewers for their constructive comments. We have provided point-by-point responses as follows.

**Reviewer #1 (Public Review):**
This study was designed to examine the bypass of Ras/Erk signaling defects that enable limited regeneration in a mouse model of hepatic regeneration. The authors show that this hepatocyte proliferation is marked by expression of CD133 by groups of cells. The CD133 appears to be located on intracellular vesicles associated with microtubules. These vesicles are loaded with mRNA. The authors conclude that the CD133 vesicles mediate an intercellular signaling pathway that supports cell proliferation. These are new observations that have broad significance to the fields of regeneration and cancer.The primary observation is that the limited regeneration observed in livers with Ras/Erk signaling defects is associated with CD133 expression by groups of cells. The functional significance of CD133 was tested using Prom1 KO mice - the data presented are convincing.The major weakness of the study is that some molecular mechanistic details are unclear - this is, in part, due to the extensive new biology that is described. Nevertheless, the data used to support some key points in this study are unclear:

We fully agree that some details of the molecular mechanisms are yet to be elucidated for the CD133+ vesicles (intercellsomes, as we named). This is the first report of a new direct cell-cell communication mechanism provoked in stress response to proliferative signal deficit.

Remarkably, many questions remain open for the molecular mechanisms for formation and functions of relatively well-characterized structures such as exosomes/EVs, despite a huge body of literature since their discoveries.

a) What is the evidence that the observed CD133 groups of cells are not due to clonal growth. Is this conclusion based on the time course (the groups appear more rapidly than proliferation) or is this based on the GFP clonal analysis?

This is indeed a very critical point for this study. Our initial thought and efforts were indeed on finding evidence that supports clonal expansion of progenitor cells. However, the experiments showed that the CD133+ cells were negative for all other stem/progenitor cell markers and that they are mature hepatocytes. CD133 expression was upregulated dramatically in regenerating livers and disappeared upon completion of liver regeneration. Furthermore, suppression of Ras-Erk signaling by Shp2 and Mek inhibitors robustly induced CD133 expression in a variety of cancer cell lines in culture in vitro.

At 2 days after PHx, we already observed big colonies, which were unlikely derived from a single initiating cell (Figure 1). The GFP clonal analysis unambiguously demonstrated the heterogenous origin of the clustered cells (Figure 3). We detected mixed GFP-positive and -negative cells within each colony, without a single colony consisting entirely of GFP-positive cells. The original colony sizes were estimated to be 10 cells or more (Figures 3G and Figure 3–figure supplement 1B). Thus, both the sizes and compositions in the GFP clonal analyses support the assertion that CD133+ cell clusters originated from multiple mature hepatocytes.

b) What is the evidence that the CD133 vesicles mediate intercellular communication. This is an exciting hypothesis, but what is the evidence that this happens? Is this inferred from IEG mRNA diversity? or some other data. Is there direct evidence of transfer - for example, the does the GFP clonal analysis show transfer of GFP that is not mediated by clonal proliferation? Moreover, since the hepatocytes are isogenic, what distinguishes the donor and recipient cells?Increased clarity concerning what is hypothesis and what is directly supported by data - would improve the presentation of this study.

Per the reviewer’s advice, we have clarified these points in the revised version. Our proposal that CD133 vesicles mediate intercellular communication was supported by these experimental results.

A). Data in Fig. 5 suggest direct trafficking of the vesicles, as CD133 existed on the filaments that bridge the tightly contacting cells. This was confirmed by two different CD133 antibodies in mouse and human. Of note, CD133+ vesicles are negative for CD9, CD63 or CD81, markers for exosomes/EVs. We could only isolate CD133+ vesicles from cell lysates in vitro and mouse tissue lysates, but not from cell supernatants from which exosomes/EVs are isolated.

B). More direct evidence of the transfer was presented in Fig. 6H, showing Myc-tagged CD133 molecules transferred from one cell to another. In response to reviewers’ comments, we now conducted correlative light and electron microscopy to characterize the exchange event around the cell-cell border at EM level (new Figure6-figure supplement 2).

C). Further experimental evidence was provided in the single and double gene KO experiments in Fig. 8E-G, suggesting the functional significance of CD133 in intercellular communication.

D). In addition to the data above, the IEG mRNA diversity analyses based on scRNA-seq support the mRNA exchange model. The isogenic CD133+ SKO hepatocytes were found to lack different IEG transcripts randomly. This is why we propose a mutually sharing model, rather than a donor and recipient model. Importantly, the mRNA diversity (entropy) model also illustrates the association of CD133 and “stemness", as described in the discussion.

In sum, we believe that a most reasonable interpretation of the current data set is a model of direct cell-cell communication via CD133+ vesicles. We take the reviewer’s point and have made changes to the text to better distinguish conclusion and hypothesis, which will be validated in future studies.

**Reviewer #2 (Public Review):**
The manuscript by Kaneko set out to understand the mechanisms underlying cell proliferation in hepatocytes lacking Shp2 signals. To do this, the authors focused on CD133 as the proliferating clusters of cells in the Shp2 knockout (SKO) livers are CD133 expressing. After excluding the contribution of progenitors that are CD133 to this cell population, the authors focused on the intrinsic regulation of CD133 by Met/Shp2 regulated Ras/Erk pathway and showed upregulation of CD133 to be a compensatory signal to overcome loss of Ras/Erk signal and suggested Wnt10a in the regulation of CD133 signal. The study then focused on the observed filament localization of CD133 in the CD133+ cluster of cells. The study went on to identify the CD133+ vesicles that contain primarily mRNA vs. microRNA like other EVs. Specifically, the authors identified several mRNA species that encode IEGs, indicating a potential role for these CD133+ vesicles in cell proliferation signal transmission to neighboring cells via delivery of the IEG mRNAs as cargos. Finally, they showed that the induction of CD133 (and by derivative, the CD133+ vesicles) are necessary for maintaining cell proliferation in the cell cluster with high proliferation capacities in the SKO livers; and in intestinal crypt organoids treated with Met inhibitors to block Ras/ERk signal.1. The identification of CD133+ vesicles is largely based on staining and costainings. Though the experiments are very well done with many controls and approaches, the authors may want to perform one or two key experiments with EM to definitively demonstrate the colocalization. For example, the mCherry experiment in Fig6H and the colocalization experiments for CD133 and HuR in Fig 7.

Many thanks for the suggestion. We now completed the two suggested key experiments with new results added to the revised manuscript. For the mCherry experiment, we conducted correlative light and electron microscopy to characterize the exchange event between cells that stably express CD133-GFP fusion protein and mCherry+ cells (new Figure 6-figure supplement 2). The CD133-GFP was clearly found in the mCherry+ cells around the border, demonstrating the intercellular traffic. For the colocalization of CD133 and HuR, we performed double immunogold staining on the isolated vesicles, with the new results presented in the revised Figure7-figure supplement 1D.

1. Since CD133+ marks the 50nM intracellsome defined by the authors, it is unclear what the CD133- vesicles used as controls are. Are they regular EVs that are larger in size? This needs better clarification as they are used as a control for many experiments such as Fig 7A.

Per the advice, we added more explanation to the revised text. We used regular EVs as the control, since they are the well-studied intercellular communication vesicles. Since the EVs are highly heterogenous, we did not choose to select a specific subpopulation of EVs. We used the well-established polymer-based precipitation method to isolate the EV fraction from cell culture supernatant for RNA-seq analysis. We did detect the enrichment of micro-RNAs in the isolated EVs, consistent with reports in the literature. Strikingly, the CD133 vesicles isolated from cell lysates showed a completely distinct RNA profile, relative to the EVs.